

# A Comparative Analysis of In-Situ Measurements of High Altitude Cirrus in the Tropics

Francesco Cairo[1], Martina Krämer[4,3], Armin Afchine[4], Luca Di Liberto[1], Sergey Khaykin[7], Lorenza Lucaferri[5], Valentin Mitev[6], Max Port[3,*], Christian Rolf[4], Marcel Snels[1], Nicole Spelten[4], Ralf Weigel[3], and Stephan Borrmann[2,3]

[1]Istituto di Scienze dell'Atmosfera e del Clima, Consiglio Nazionale delle Ricerche, Rome, Italy
[2]Particle Chemistry Department, Max Planck Institute for Chemistry, Mainz, Germany
[3]Institute for Atmospheric Physics, Johannes Gutenberg University, Mainz, Germany
[4]Institute of Energy and Climate Research – IEK7, Forschungszentrum Jülich, Jülich, Germany
[5]Dipartimento di Fisica, Università degli Studi di Roma "Tor Vergata", Rome, Italy
[6]Centre Suisse d'Electronique et de Microtechnique, Neuchâtel, Switzerland
[7]Laboratoire Atmosphères, Observations Spatiales (LATMOS), UVSQ, Sorbonne Université, CNRS, IPSL, Guyancourt, France
[*]now at: Montessori Zentrum Hofheim, Hofheim, Germany

**Correspondence:** Francesco Cairo (f.cairo@isac.cnr.it)

**Abstract.** We analyze cirrus cloud measurements from two dual-instrument cloud spectrometers, two hygrometers and a backscattersonde in view to connect cirrus optical parameters usually accessible by remote sensing with microphysical size resolved and bulk properties accessible in situ. Specifically, we compare the particle backscattering coefficient and depolarization ratio to the particle size distribution, effective and mean radius, surface area density, particle aspherical fraction and ice water

content. Data have been acquired by instruments on board the M55 Geophysica research aircraft during July and August 2017 during the Asian Monsoon campaign based in Kathmandu, Nepal, in the framework of the StratoClim (Stratospheric and upper tropospheric processes for better climate predictions) project. Cirrus have been observed over the Hymalaian region between 10 km and the tropopause, situated at 17-18 km. The observed particle number densities varied between 10 and $10^{-4}$ $cm^{-3}$ in the dimensional range from 1.5 to 468.5 $\mu$m in radius. Correspondingly, backscatter ratios from one tenth up to 50 have been

observed.

Optical scattering theory has been used to compare the backscattering coefficient computed from measured particle size distribution with those directly observed by the backscattersonde. The aspect ratio of the particles, modeled as spheroids for the T-matrix approach, was left as a free parameter to match the calculations to the optical measures. The computed backscattering coefficient can be set in good agreement with the observed one, but the match between simulated and determined depolarization

ratios is insufficient, however. Relationships between ice particle concentration, mean and effective radius, surface area density and ice water content with the measured backscattering coefficient are investigated for an estimate of the bulk microphysical parameters of cirrus clouds from remote sensing lidar data. The comparison between particle depolarization and aspherical fraction as measured by one of the cloud spectrometers equipped with a detector for polarization, represents a novelty since it was the first time the two instruments are operated simultaneously on aircraft. The analysis shows the difficulty of establishing





an univocal link between depolarization values and the presence and amount of aspherical scatterers. This suggests the need
of further investigation that could take into consideration not only the fraction of aspheric particles but also their predominant
morphology.

## 1 Introduction

Cirrus are clouds composed of micron to millimeter sized ice crystals of various shapes that form in the upper troposphere
between 4 and 20 km above sea level (Lynch et al., 2002). Cirrus higher elevations are usually found in the tropics, where
their highest occurrence frequency is also recorded, and lower elevations are found in polar regions (Sassen et al., 2008, 2009).
Tropical cirrus originate either from outflows from deep convective clouds (liquid-origin cirrus) or from vertical uplifting of
air (in situ-origin cirrus) associated with Kelvin or gravity waves as well as with the synoptic-scale tropospheric tropical ascent
(Jensen et al., 1996; Pfister et al., 2001; Immler et al., 2008; Fujiwara et al., 2009; He et al., 2012; Krämer et al., 2016; Luebke
et al., 2016; Wernli et al., 2016). Studies of these clouds are important for a better understanding of their impact on the climate
as they play a crucial role in earth's radiation budget (Prabhakara et al., 1993; Campbell et al., 2016; Lolli, 2017; Krämer et al.,
2020). Their impact is based on two effects (Stephens, 2002, 2005): i. a greenhouse potential that traps the outgoing long-wave
radiation emitted by the earth and the atmosphere underneath; ii. their albedo that reflects the incoming solar radiation. The
balance between the cirrus induced warming and cooling depends on their coverage, height, thickness, horizontal and vertical
temperature distribution as well as ice crystal size and shape distributions within the clouds (Lynch, 1996; Boucher et al.,
2013). Moreover, cirrus are an essential modulator of the water budget in the upper troposphere and in the stratosphere (Luo
et al., 2003; MacKenzie et al., 2006; Corti et al., 2008; Fueglistaler et al., 2009).

The qualitative and quantitative assessment of the cirrus properties over large spatial and temporal scales require the use of
satellite data. Methods exist to provide spaceborne retrievals of cirrus bulk and microphysical parameters. Passive visible or
thermal, spectrally resolved measurements have been used to infer the cloud optical depth, ice particle effective radius ($R_{eff}$),
Ice Water Path (Meyer and Platnick, 2010; Sourdeval et al., 2013; Guignard et al., 2012; Sourdeval et al., 2015) but may
subsample clouds of small optical thickness. Active remote sensing by radars and lidars can be more sensitive to thin clouds and
can still provide information on cirrus geometrical and optical properties with high spatial and temporal resolutions. Vertical
profiles of cirrus extinction, Ice Water Content (IWC) and $R_{eff}$ are retrieved using lidar and/or radar measurements (Austin
et al., 2009; Delanoë and Hogan, 2010). Global mapping of cirrus properties is obtained from satellite borne instruments like
the CALIOP (Cloud and Aerosol Lidar with Orthogonal Polarization) lidar aboard the CALIPSO (Cloud-Aerosol Lidar and
Infrared Pathfinder Satellite Observation) polar orbiting satellite (Nazaryan et al., 2008). Such measurements have begun to
include estimates of ice crystal number concentrations ($N_{ice}$). In fact, climate model parameterizations of ice cloud optical
properties are based on $R_{eff}$ and IWC (Fu, 2007) but these two do not fully constrain the ice cloud Particle Size Distribution



(PSDs) and its optical properties (Mitchell et al., 2011). Moreover, the knowledge of $N_{ice}$ on a global scale would ameliorate the understanding of ice nucleation and its parametrization in climate models. A refinement of the ice crystal nucleation rates would in turn improve the predictions of $R_{eff}$. Mitchell et al. (2018) use co-located observations from the Infrared Imaging Radiometer (IIR) and from CALIOP to retrieve $N_{ice}$, $R_{eff}$ and IWC in semi-transparent cirrus clouds, while Sourdeval et al.
(2018) employ combined lidar–radar measurement to provide satellite estimates of $N_{ice}$ using a methodology that constrains moments of a parameterized Particle Size Distribution (PSD) through lidar extinction and radar reflectivity.

Cirrus clouds' microphysical properties have been characterized by in-situ measurements taken during several airborne observation campaigns (Thomas et al., 2002; Schiller et al., 2008; Krämer et al., 2009; Frey et al., 2011; Luebke et al., 2013; Frey et al., 2014; Krämer et al., 2016; Schumann et al., 2017). A review of these studies and of the challenges they present is
reported in Baumgardner et al. (2017) while Krämer et al. (2020) describe extensive statistics of meteorological parameters, IWC, $N_{ice}$, ice crystal mean radii ($R_{mean}$), relative humidities with respect to ice and water vapor mixing ratios from airborne in situ measurements performed during 150 flights in 24 campaigns, in midlatitudes and the tropics. Such observational activity, in addition to being essential for shedding light on the processes of formation, aging and dissipation of the clouds, is of great help in the interpretation of satellite sensor data, allowing the calibration and validation of the retrievals of cirrus microphysical
and bulk parameters by comparing them with corresponding results of in situ measurements.

In the present work we intend to compare optical measurements of cirrus clouds, i.e. particle backscatter coefficient $\beta$ and total particle depolarization $\delta_{TA}$ with bulk and microphysical parameters observed by cloud spectrometers and hygrometers. The optical measurements are generally accessible to lidar probing, but in our case have been taken in situ by a backscattersonde, henceforth they are directly comparable with the other data acquired in situ as all measurements originate from the same air
parcel. Measurements have been taken during the aircraft field campaign of the EU funded project StratoClim (Stratospheric and upper tropospheric processes for better climate predictions) carried out in South Asia in 2017. A full description of the campaign is provided by Stroh et al. (2022, same issue). The South Asian campaign of the high altitude research aircraft M55 Geophysica (Stefanutti et al., 1999) focused on detailed observations of atmospheric transport and physical-chemical processes which dominate the input of air and aerosols into the (sub-)tropical stratosphere. In the present work we make use of data from
seven flights, performed on 29 and 31 July and on 2, 4, 6, 8 and 10 August 2017. During these flights the airplane penetrated into cirrus clouds several times for nearly seven hours of observations in clouds over approximatively 35 flight hours.

We verify the ability and limits of optical modeling in reproducing the results of remote sensing optical measurements from the concomitant in-situ microphysical ones, thus assessing the compatibility of the two datasets.

Closure studies between aerosol light scattering coefficients and aerosol PSD with optical modeling are common in atmo-
spheric science, and they often use measurements from in situ nephelometers, and Mie theory applied to cloud spectrometers data (Wex et al., 2002; Teri et al., 2022). Similar approaches have been also attempted by comparing lidar or backscatter probe measurements with cloud spectrometers mounted on balloons or on airplanes, in the characterization of polar stratospheric clouds (Schreiner et al., 2002; Deshler et al., 2000; Scarchilli et al., 2005; Snels et al., 2021; Cairo et al., 2023). Comparatively fewer studies are present in comparing the size distributions of ice crystals in cirrus clouds, and their modeled and measured
optical scattering properties, this is particularly true for backscattering measurements (Wagner and Delene, 2022). The reason





probably lies in the greater difficulty to obtain significant and definitive confirmations from such optical closures. This is likely because of the uncertainties affecting both from the particle size distribution measurements and their optical modeling, arising from the large dimensional range of the ice crystals extending for more than three orders of magnitude, and the great variety of shapes that they can take simultaneously within the cloud (Bailey and Hallett, 2009). A further complication arises from the presence of rough irregular surfaces, corners and edges in ice crystals (Schnaiter et al., 2011). Actually, modeling of the optical properties of cirrus clouds is a formidable problem with probably not always a valid solution, despite the numerous methods available for calculating scattering from aspherical particles (Mishchenko et al., 1999; Konoshonkin et al., 2017a). In fact many light-scattering computation methods have been employed to calculate the scattering properties of cirrus particles and we may quote the finite-difference time-domain (FDTD) method (Sun et al., 1999), the discrete dipole approximation (DDA) (Yurkin et al., 2007), the boundary element method (Groth et al., 2015), the pseudo-spectral time-domain method (Liu et al., 2012), the surface-integral equation method (Nakajima et al., 2009), and geometrical optics in various implementations: improved geometric optics method (IGOM) (Yang and Liou, 1996a), geometric optics integral equation (GOIE) (Ishimoto et al., 2013) and ray-tracing geometric optics method (GOM) (Macke et al., 1996). The T-matrix theory offers solution to the computation of electromagnetic scattering from axisymmetric particles and has practical advantages over other methods, largely due to its analytical character and the exploitation of particle symmetries, which considerably simplifies the calculation, and has long been used to study the scattering properties of cirrus clouds. Mishchenko et al. (1997) used it to compute the backscattering from horizontally oriented ice platelets in cirrus, Baran et al. (2001) modeled absorption and extinction properties of the finite hexagonal ice columns and plates in random and preferred orientation, Liu et al. (2006) reported results on the scattering properties of small cirrus crystals modeled as mixtures of polydisperse, randomly oriented spheroids and cylinders with varying aspect ratios, Bi and Yang (2014) employed it to compute the optical properties of randomly oriented ice crystals of various shapes including hexagonal columns, hollow columns, droxtals, bullet rosettes and aggregates. T- matrix theory has also been used to simulate the response of cloud spectrometers with forward scattering geometries, when exposed to clouds of aspherical particles (Borrmann and Luo, 2000).

Finally, the possibility of using microphysical observations taken along with optical observations usually accessible in remote sensing but here obtained with quasi in situ techniques (the air mass sampled by the backscattersonde is for all practical purposes the same sampled by hygrometers and cloud spectrometers) allows us establish relationships between $\beta$ and $\delta_{TA}$ and cloud parameters retrieved from the measured PSDs, as $N_{ice}$, $R_{mean}$, $R_{eff}$, Surface Area Density (SAD), particle Aspherical Fraction (AF), and cloud IWC, this latter retrieved from a comparison of data from hygrometers that differentially measure the proportion of gaseous and total water. These empirical relationships are useful in interpreting lidar measurements of cirrus clouds in terms of their microphysical and bulk parameters.

## 2 Instruments and Data

We have compared and used data from the two dual instrument cloud spectrometers that flew aboard the aircraft Geophysica, the NIXE-CAPS and the CCP. The CCP and NIXE-CAPS instruments (see Section 2.2) are both developments by Droplet



Measurement Technologies (DMT) and are nearly identical in their optical measurement technology, the sensors used for pressure and temperature, and the flow measurement technology (Prandl's pitot tube system). The IWC has been derived from the total water hygrometer FISH and the water vapor hygrometer FLASH (see Section 2.3); the cloud optical parameters have been derived from the backscattersonde MAS (see Section 2.1) which measures in situ particle backscattering coefficient and depolarization ratio at 532 nm. Data from MAS are also compared on some cases of interest with the 532 nm elastic lidar MAL on board the same aircraft.

## 2.1 Particle backscatter measurements: MAS and MAL instruments

### 2.1.1 The MAS backscattersonde

Optical observations have been provided by the backscattersonde MAS, located in a bay beneath the pilot's cockpit, facing sideways on the right. It emits polarized laser light at 532 nmm and collects the light backscattered from the portion of atmosphere in close proximity ($3$–$10\ m$) to the instrument, sensitive to optically detectable (i.e. whose diameter is greater than few tenths of $\mu m$) cloud particles and aerosols. Polarization resolved observations allow to detect the particle's asphericity, hence thermodynamic phase. The instrument is basically a polarization diversity Rayleigh lidar system that measures in-situ the same atmospheric parameters which are accessible to lidars. The sampling volume is approximately $10^{-3}\ m^{-3}$, the resolution is $10\ s$, corresponding to a 1.5-1.9 $km$ horizontal resolution along the aircraft trajectory, given the $154 \pm 39\ m\ s^{-1}$ aircraft speed at altitude (Weigel et al., 2021a, b).

The backscattered light is split according to parallel and perpendicular polarization with respect to the linearly polarized laser light, allowing the measurement of the Backscatter Ratio BR, the Volume Depolarization $\delta$ and the Total Aerosol Depolarization $\delta_{TA}$. These optical parameters follows the usual definitions (Cairo et al., 1999) reported hereby for convenience; in the following the subscripts $mol$ and $A$ denote respectively the molecular and aerosols contribution to the optical coefficients, and $cross$ and $par$ denote the perpendicular and parallel polarization of the backscattering coefficient $\beta$ (Collis and Russell, 1976).

$$BR = \frac{\beta_A^{cross} + \beta_{mol}^{cross} + \beta_A^{par} + \beta_{mol}^{par}}{\beta_{mol}^{cross} + \beta_{mol}^{par}} \tag{1}$$

$$\delta_T = \frac{\beta_{mol}^{cross} + \beta_A^{cross}}{\beta_{mol}^{par} + \beta_A^{par} + \beta_{mol}^{cross} + \beta_A^{cross}} \tag{2}$$

$$\delta_A = \frac{\beta_A^{cross}}{\beta_A^{par}} \tag{3}$$

$$\delta_{TA} = \frac{\beta_A^{cross}}{\beta_A^{cross} + \beta_A^{par}} \tag{4}$$

Note the different definitions of Particle Depolarization $\delta_A$ and Total Particle Depolarization $\delta_{TA}$ which will be used both in the following. Formulae to pass from one to the other definition can be found in Cairo et al. (1999).

The signal detected by the backscattersonde channels is directly proportional to $\beta_A^{par(cross)} + \beta_{mol}^{par(cross)}$. The backscatter ratio BR is derived by a calibration procedure that uses concomitant measurements of pressure and temperature to retrieve





the air density, and defines a suitable constant K – taking into account the molecular scattering cross section as well as the
instrumental sensitivity - in order to achieve BR=1 from the signal detected in air masses where no particles are present. A
description of the instrument and data processing can be found in Cairo et al. (2011).

Figure 1 reports the 2D histogram of frequency distribution of BR with respect to geometrical altitude; the particle obser-
vations clusters in two regions, namely between 7.5 and 10 km, and above 13 km up to the tropopause region, the Cold Point
Tropopause being at 17 km on average. In the following we will focus on observations taken in that upper region. We identify
a cloud presence when BR>1.2.

Figure 2 exhibits $\delta_{TA}$ with respect to temperature, for clouds observed above 11 km. Total Particle Depolarization ranges
from 20 to 50%. In the upper part of the troposphere, i.e. when temperatures fall below 200K, a negative trend with respect
to temperature can be discerned; in that altitude range, an increase of $\delta_{TA}$ with altitude as has been often reported for cirrus
(Sassen and Benson, 2001; Sunilkumar and Parameswaran, 2005; Cairo et al., 2021). However, at higher temperatures, namely
at around 205K and 215K, observations of highly depolarizing clouds are also here reported.

### 2.1.2 The MAL lidars

The lidars MAL1 and MAL2 (Miniature Aerosol Lidars Mk1 and Mk2) are elastic backscatter-depolarization instruments,
operating at 532 nm (Martucci et al., 2005). They operate with micropulse lasers, having pulse repetition rates of 4.5-5.5 kHz
and are packed each in a single pressurized box. MAL1 is installed on M55 Geophysica for upward probing, while MAL2 is
installed for downward probing. The measured values are the aerosol backscatter ratio and volume depolarization ratio. From
the measured signals it is possible to obtain the aerosol backscattering coefficient and depolarisation ratio. The definition of the
measured and obtained values follows eqs. (1-4). The high-repetition rate combined with low pulse energy of the laser gives
the possibility to use a photon-counting detection/acquisition system with high dynamic range. This makes possible to start the
lidar profile as close to the aircraft, as allowed by the geometrical overlap function, i.e. at 400 m from the platform delivering
backscatter and depolarization profiles every minute, with a vertical resolution of 50m. The processing procedure is based on
comparison of the signal from the clouds with the molecular backscatter signal from aerosol-free parts of the lidar profile.
In this it is similar to the one for MAS. The difference with MAS is that in MAL the aerosol free atmospheric volumes are
identified in the lidar profile, respectively at some range above and below the aircraft, and not at the flight altitude. The lidars
participated in a number of campaigns with M55 Geophysica (Cairo et al., 2004; Molleker et al., 2014; Mahnke et al., 2021).
This includes a comparison case with CALIPSO lidar CALIOP during RECONCILE campaign (Mitev et al., 2012).

## 2.2 Cloud spectrometers

### 2.2.1 The NIXE–CAPS instrument

The NIXE-CAPS (for details see Luebke et al. (2016); Costa et al. (2017)) is located below the aircraft wing and incorporates
two instruments: the CAS–DPOL (Cloud and Aerosol Spectrometer with detector for polarization) and the NIXE–CIPgs (Cloud
Imaging Probe grey scale); it also includes an air speed sensor and a temperature probe. CAS-DPOL is a light-scattering probe



covering the particle size range of 0.3 to 25 $\mu m$ in radius. Moreover, the CAS records the change of polarization in the backward-scattered light, thus giving information about the particle asphericity.

NIXE–CIPgs is an optical array probe that covers the particle size range between 7.5 and 468.5 $\mu m$. The instrument captures the image of a cloud particle by using a 64-element photodiode array (15 $\mu$m resolution) to generate two-dimensional shadow

185 images that can be analyzed for particle size and asphericity using various algorithms (Costa et al., 2017).

For aircraft speeds between 240 and 80 $m\ s^{-1}$, the instruments' sampling volumes limit the particle concentration measurements to concentrations above 0.02 to 0.05 $cm^{-3}$ (NIXE–CAS–DPOL) and about 0.0001 to 0.001 $cm^{-3}$ (NIXE–CIPgs).

The particle size distributions (PSDs) of CAS-DPOL and CIPgs are merged into a single PSD covering the range of 0.3 to 468.5 $\mu m$, where the size bins up to 20 $\mu m$ are taken from the CAS-DPOL and those larger than 20 $\mu m$ from the CIPgs. This

190 threshold is used since the CIPgs has a larger sampling volume than the CAS-DPOL, thus providing better particle sampling statistics. Particles larger than 1.5 $\mu m$ in radius are classified as cloud, while the smaller particles are considered aerosols.

The PSDs are reported for each second. In the present work the PSDs have been averaged over $10s$ and synchronized to the backscattersonde data. In the following, in addition to the PSD, we have used the "combined" aspherical fraction $AF_i$ which refers to fraction of aspherical particles detected in size channels both from NIXE-CAS and NIXE-CIPgs. The AFs are derived

195 as described by Costa et al. (2017)

The lidar-based aerosol depolarization, being an optical measure, may be biased toward a particular size range of the PSD, depending on the respective particle number densities and absolute sizes, as many smaller ones can be as "active" as few bigger ones. In order to study its relationship with the microphysical measurements, we have used both the ordinary mean $AF$ defined as:

200
$$AF = \frac{\sum_{i=1}^{max} Af_i n_i}{\sum_{i=1}^{max} n_i} \tag{5}$$

and an "effective" mean $AF_{eff}$ where in the mean the $AF_i$, detected in the single size bin $i$ where $n_i$ particles with radii between $r_{i-1}$ and $g_i$ have been counted, are weighted proportionally to the mean cross sectional area $S_i$:

$$S_i = \pi \left( \frac{r_{i-1} + r_i}{2} \right)^2 \tag{6}$$

relative to that bin.

205
$$AF_{eff} = \frac{\sum_{i=1}^{max} Af_i S_i n_i}{\sum_{i=1} S_i n_i} \tag{7}$$

In this way the $AF_i$ are weighted not only with the numerical density of the particles in the bin $i$, but also in proportion to a proxy of their scattering cross section. In the subsequent analysis, however, we did not find notable differences when seeking a correlation between the observed depolarization and $AF$ or $AF_{eff}$. Hence in the following we will report only the results relating $AF$.





### 2.2.2 The CCP instrument

The CCP (Cloud Combination Probe) combines a CDP (Cloud Droplet Probe) with a CIPgs (Cloud Imaging Probe with grey scale), whose measurement technique as well as the characteristics of the measurement data analysis have already been described in detail (Frey et al., 2011; Molleker et al., 2014; Klingebiel et al., 2015; Grulich et al., 2021). The CCP-CDP is a light scattering probe comparable to the CAS-Dpol (or NIXE-CAS), but it covers the range of 2.5-46 $\mu$m in particle diameter with a size resolution of 1 - 2 $\mu$m (Mei et al., 2020), encompassing also the uppermost range of the aerosol's size spectrum. The sampling area of the CCP-CIPgs was examined by Klingebiel et al. (2015), and this analysis yielded 0.27±0.025 $mm^2$ with an uncertainty of less than 10%. In contrast, the CCP-CIPgs is an optical array probe designed to detect cloud particles and hydro meteors with a resolution of 15 $\mu$m. CIPgs captures images of cloud elements using a 64-element photodiode array to obtain two-dimensional images with nominal detection diameters ranging from 15 to 960 $\mu$m. CCP-CDP and NIXE-CAS performances have been frequently validated by glass sphere calibrations. Before or after each flight, CCP-CIPgs and NIXE-CIP calibrations were performed using spinning disks carrying opaque spots sized to the particle range to be detected. Particle concentration data measured with CCP are corrected for compression under measurement conditions using a thermodynamic approach developed by Weigel et al. (2016).

### 2.3 FISH and FLASH Hygrometers

The FISH (Fast In situ Stratospheric Hygrometer) instrument is a closed-path Lyman-$\alpha$ photofragment fluorescence hygrometer (Zöger et al., 1999; Meyer et al., 2015) used to measure $H_2O_{tot}$ in the range of 1–1000 ppmv between 50 and 500 hPa with an accuracy and precision during StratoClim of 7% and 0.3 ppmv. FISH is calibrated versus a reference frost point mirror MBW 373 LX on ground before and after each flight to ensure high data quality. On board Geophysica, the inlet for the $H_2O_{tot}$ hygrometer FISH is mounted on the side of the aircraft, heated and with a 90° bend to quickly evaporate ice crystals. In the present work the original $1s$ resolution data have been averaged over $10s$.

FLASH (FLuorescent Airborne Stratospheric Hygrometer; for details see Khaykin et al. (2013, 2022)) also uses the Lyman-$\alpha$ photofragment fluorescence technique for the detection of water vapor, but its inlet is designed to sample only the gas phase. The detection range is 1–1000 ppmv with an accuracy and precision of <9% and 0.5 ppmv. The time resolution is 1 Hz but in the present work data have been averaged over $10s$. The $H_2O_{gas}$ hygrometer FLASH is mounted below the wing and equipped with its own inlet.

Clear-sky data from the two hygrometers have been inter-compared together and with those from a third hygrometer on board the M55-Geophysica, ChiWIS (Chicago Water Isotope) designed for airborne measurements of vapour phase water isotopologues in the dry Upper Troposphere - Lower Stratosphere with integrated cavity output absorption spectroscopy (Sarkozy et al., 2020). The comparison showed excellent agreement between these in situ instruments (Singer et al., 2022).



## 3  Methods

We first compare the particle backscattering coefficient derived by computing it from the measured PSD, namely $\beta_{NIXE-CAPS}$ with the $\beta$ measured by the backscattersonde MAS. We then present regressions between particle backscattering coefficient $\beta$ and particle number density $N_{ice}$, mean radius $R_{mean}$, effective radius $R_{eff}$, SAD and IWC. We will assume that the dispersion of the measurements among the various instruments compared could be used as an estimate of the uncertainty to be

attributed to such regressions. In addition, we investigate the relation between the particle aspherical fraction (AF) with the measured total particle depolarization $\delta_{TA}$ and with the other cloud microphysical and environmental parameters.

### 3.1  Optical Modelling

The backscattering coefficient and depolarization ratio have been computed with the GRASP (Generalized Retrieval of Aerosol and Surface Properties) Spheroid Package coupled with Mie scattering computations performed using a code available from

the NASA's OceanColor Web site ($https : //oceancolor.gsfc.nasa.gov/docs/ocssw/bhmie_8py_source.html$), one of the python's avatars of the Bohren and Huffman fortran code originally published in their book on light scattering (Bohren and Huffman, 2008).

GRASP is the first unified algorithm developed for characterizing atmospheric properties gathered from a variety of remote sensing observations (Dubovik et al., 2014), whose software packages are available on the Web project repository ($https :$

$//www.grasp-open.com/$). The Spheroid Package allows fast, fairly accurate and flexible modeling of single scattering properties by randomly oriented spheroids with aspect ratio spanning from 0.3 to 3. The details of the scientific concept are described in the paper by Dubovik et al. (2006). The code uses kernel look-up tables including results of calculations using T-Matrix codes for particle size parameters where convergence was acquired, and geometric-optics-integral-equation code (Yang and Liou, 1996b, 2006) for greater size parameters where T-matrix codes did not converge. The two methods have been

shown to produce comparable results over the size range in which both are applicable (Dubovik et al., 2006). The GRASP spheroid package thus provide backscattering coefficients for randomly oriented spheroids with Aspect Ratios (AR) from $\sim$ 0.3 (flattened spheroids) to $\sim$ 3.0 (elongated spheroids) and covering size parameters from $\sim 0.01$ to $\sim 517$ (when a wavelength of $532nm$ is used) for a wide range of the particle complex refractive index, the real part form 1.3 to

1.6, the imaginary part from 0.0005

to

0.5. Since in our case the size parameters of the particles detected by the cloud spectrometers extend for more than an order of magnitude beyond the GRASP limit, we were forced to extrapolate the GRASP results using an approximation that makes use of the Mie code, even for aspherical particles.

To justify this extrapolation, we recall some characteristics of the backscattering and of the depolarization from aspherical

scatterers, depending on their size. According to T-Matrix computations on randomly oriented spheroids and cylinders, depolarization is negligible when their size parameters less than unity (given the wavelength used in our study, this corresponds to equivalent radius around 0.1 $\mu$m) Then it grows to a maximum for size parameters of the order of ten (equivalent radius





around 1 $\mu$m), to decrease to an asymptotic value which stabilizes for size parameters approximately greater than one hundred (particle radius greater than 10 $\mu$m). The maximum and the asymptotic values of the depolarization vary in dependence of the
specific AR. The variability of these values are unrelated, and there is no general relationship that links the peak and asymptotic depolarization values to the AR of the spheroids.

For what concerns the backscattering, when the particle size parameter is below unity the T-matrix backscattering efficiency reproduces the Mie results for surface equivalent spheres, then reaches a minimum as the size parameter increases, to go up again and stabilize, at a few tens of size parameters, at values in a constant relationship with Mie's calculations. This constant
is often less than unity; it can exceed the unit by a few decimal places in a few cases, for ARs around 0.5. (Mishchenko et al., 1996, 2002). The increase in backscattering for AR around 0.5, probably due to a sort of mirror reflection effect that predominates with respect to the backscattering of spheres of equivalent radius, does not exceed a factor of 1.5-2. Conversely the backscattering depression, which is a more general feature of aspherical scatterers, can be as large as a factor 10 or more. The dependence of the single particle's depolarization on its shape and size has been studied extensively by Liu and Mishchenko
285 (2001).

The GRASP calculations extend well into the region of asymptotic values for the depolarization and for the spherical vs aspherical ratio of backscattering, so that we can extrapolate its results with confidence. Beyond the computational limits of GRASP we have in fact set the depolarization of the particles as constant, i.e. equal to its asymptotic value. Moreover we have used the Mie code for the calculation of the backscattering efficiencies, suitably rescaled by a constant factor so as to make
the scaled Mie backscattering to overlap with the asymptotic values calculated by GRASP. Such constant factor was calculated from the ratio of Mie vs T-matrix efficiencies in the radius dimensional region from 5 to 14 $\mu$m (60-180 size parameters). It turned out to be always less than unity.

We thus calculated, for 25 different Aspect Ratios from 0.3 to 3, the backscattering efficiencies $Q_l^{AR}$ and the depolarizations $\delta_l^{AR}$ on a grid of 2000 radius points $R_l$ equally spaced on a logarithmic scale and extending from 0.005 to 1000 $\mu$m. We
used the GRASP computations for radii below 14 $\mu$m and we extrapolated these result with the Mie code for larger radii, as outlined above. We used the value of 1.31 as refractive index relative to ice. To get rid of the oscillating nature of the backscattering efficiency, for each radius $R_l$ the values of $Q_l^{AR}$ and $\delta_l^{AR}$ were actually obtained as averages over a narrow lognormal distribution centered at $R_l$, with variance 1.01. Such averages were computed over the lognormal with a finer subgrid of 500 points $R_k$, equally spaced on a logarithmic scale and extending over 1-$\sigma$ from the lognormal center .
Figure A1 and A2 in the Appendix reports the results of such extrapolation. In those figures, particle backscattering coefficients and depolarization ratio have been calculated for a reference particle density of 1 cm$^{-3}$, for different AR, and displayed for particle radius from 0.01 to 100 $\mu$m.

These efficiencies were then used to calculate the backscattering coefficients associated with the PSD measurements. For each PSD size bin i, we computed the arithmetic average of the $M_i$ lattice radius points falling within the size channel limit





and the corresponding average efficiency and depolarization

$$r_i = \frac{\sum_{l=1}^{M_i} R_l}{M_i} \tag{8}$$

$$Q_i^{AR} = \frac{\sum_{l=1}^{M_i} Q_l^{AR}}{M_i} \tag{9}$$

$$\delta_{Ai}^{AR} = \frac{\sum_{l=1}^{M_i} \delta_{Al}^{AR}}{M_i} \tag{10}$$

then we used the concentration of particles $n_i$ in the size bin $i$ to derive the contribution of that bin to the backscattering

coefficient and the depolarization, and summed over the bins. In the case of depolarization, the average was weighted with the

backscattering coefficient of the same channel. This was repeated for all the 25 AR.

$$\beta_i^{AR} = n_i \pi r_i^2 Q_i^{AR} \tag{11}$$

$$\beta_{NC}^{AR} = \sum_{i=1}^{max} \beta_i^{AR} \tag{12}$$

$$\delta_{ANC}^{AR} = \frac{\sum_{i=1}^{max} \delta_i^{AR} \beta_i^{AR}}{\beta_{NC}^{AR}} \tag{13}$$

Uncertainties in the backscattering coefficient $\Delta(\beta_{NC}^{AR})$ and $\Delta(\delta_{ANC}^{AR})$ due to sizing and counting errors were derived using

a weighted error propagation in quadrature method (Berendsen, 2011):

$$\Delta(\beta_{NC}^{AR}) = \sum_{i=1}^{max} \sqrt{\left(\pi r_i^2 Q_i^{AR} \Delta(n_i)\right)^2 + \left(2\pi r_i Q_i^{AR} \Delta(r_i)\right)^2} \tag{14}$$

$$\Delta(\delta_{ANC}^{AR}) = \frac{1}{\beta_{NC}^{AR}} \left( \sum_{i=1}^{max} \Delta_i^{AR} \sqrt{\left(\pi r_i^2 Q_i^{AR} \Delta(n_i)\right)^2 + \left(2\pi r_i Q_i^{AR} \Delta(r_i)\right)^2} + \delta_{NC}^{AR} \Delta(\beta_{NC}^{AR}) \right) \tag{15}$$

where $\Delta(n_i)$ is the uncertainty in concentration following Poisson statistics and $\Delta(r_i)$ is the uncertainty in particle radius for

channel $i$, (Horvath et al., 1990, Baumgardner et al. 2014) here one half the channel width has been used as radius uncertainty.

(Wagner and Delene, 2022).

$$\Delta(n_i) = \frac{\sqrt{n_i}}{n_i} \tag{16}$$

$$\Delta(r_i) = \frac{\Delta r_i}{2} \tag{17}$$

To perform the optical computations we use the complete dimensional range of NIXE-CAPS particle detectability, which

extends lower down to 0.3 $\mu$m in radius. This was done because the measurement of $\beta$ made by the backscattersonde is sensitive

to the backscattering from all the particles present in the sampled air mass. An estimation of the contribution of particles in the

0.3-1.5 range can be given by inspecting Figure A7 in the Appendix, where the occurrence histograms of the computation of $\beta$





for NIXE-CAPS measuring from 0.3 $\mu$m (vertical axis) or 1.5$\mu$m (horizontal axis) is reported. Such contribution is always low and negligible, except in some cases at medium-to-low values of the $\beta$, where neglecting the 0.3-1.5 1.5$\mu$m part of the PSD

can lead to a detectable underestimation of the $\beta_{NC}$, roughly of a factor 2.

## 3.2 Cirrus cloud microphysical parameters

For the comparison with the measured backscatter signals, we exclude the aerosol component of the particulate, by setting 1.5 $\mu$m in radius as a lower limit for the cloud particle radius. We calculate the cloud particle concentration $N_{ice}$, mean mass radius $R_{mean}$ (Krämer et al., 2009) calculated from IWC/$N_{ice}$ – the particles assumed as spheres, effective radius $R_{eff}$ defined as in

(18) (Schumann et al., 2011) where the definition on the second equality applies only to spherical particles and has not been used in the present work,

$$R_{eff} = \frac{3VD}{4SAD} = \frac{\int r^3 PSD(r)\,dr}{\int r^2 PSD(r)\,dr} \tag{18}$$

and SAD, VD stands for, respectively, Surface Area Density and Volume Density. To compute SAD and VD we have used the the m–D relation described in Krämer et al. (2016). To this end, the PSD were averaged over 10 s. To retrieve cirrus cloud IWC

the FLASH water vapor measurements were subtracted from the FISH total water data, once corrected for inlet enhancements (see Afchine et al. (2018)). This was done for 5 out of the 8 flights under assessment. For two flights (on 31 July and 4 August) when FISH was not operating, the IWC was computed from NIXE-CAPS particle volumes with an estimate of the ice density of 0.92 g cm $^{-3}$. An extensive assessment of the methodology to retrieve IWC from the two hygrometers and from the PSDs, as well as a comparison between the two approaches, can be found in Afchine et al. (2018) who demonstrated the equivalence

of the two methods.

## 4 Results

### 4.1 Cloud Spectrometers Comparison

We have first compared the PSD from the two cloud spectrometers in terms of $N_{ice}$, $R_{mean}$, SAD and VD, these latter evaluated in terms of spherical particle approximation. The particle backscattering from the PSD, namely $\beta_{NIXE-CAPS}$, and

$\beta_{CCP}$ were computed based on Mie theory and subsequently compared.

The NIXE-CAPS $N_{ice}$ resulted to be lower, of roughly a factor 2, than the corresponding one measured by CCP in the low to medium particle concentration regime, while the two determinations resulted to be more on a 1:1 line in the high particle concentration regime. Conversely NIXE-CAPS $R_{mean}$ resulted larger than the corresponding from what measured by CCP in the mid to high $R_{mean}$ regime, of roughly a factor 2 (see Figures A3 and A4 in Appendix). With NIXE-CAPS measuring

larger and, at low concentration, fewer particles, and wider PSD as well, not surprisingly we found a slight mismatch also in surfaces and volumes, with NIXE-CAPS measuring SAD and VD a factor 2 larger on the average than the corresponding CCP acquisitions.

The comparison of the the particle backscattering $\beta_{NIXE-CAPS}$, and $\beta_{CCP}$ computed from the PSD by use of Mie theory again produced $\beta_{NIXE-CAPS}$ larger than the corresponding $\beta_{CCP}$ by roughly a factor 2 (see figure A5 in Appendix).

An in-depth analysis of the datasets from the two cloud spectrometers showed that in the common size range of 1.25 - 15 $\mu$m particle radius, CAS and CDP agreed very well with each other. Conversely, an offset was found between NIXE-CAPS-CIPgs and CCP-CIPgs-data which was not a product of different evaluation methods or filter criteria of the image files but was very likely due to a hardware problem. A failure of the lasers' temperature stabilization was identified to lead to a loss of beam intensity at higher altitudes and therefore a less illuminated sample volume and diode array of the CCP-CIPgs. This

resulted in a slightly lower instrument particle detection sensitivity which could be only identified by a comparison of particle habits like size and number concentrations in the range of both CIPgs instruments measured at significantly elevated altitudes (Port, 2021). An implication of this finding on a similar comparison of the same two instruments from previous simultaneous measurements at lower altitudes (Mei et al., 2020) can be ruled out. Therefore for all further analyses of the optical and microphysical parameters, only the NIXE-CAPS data set was used.

## 370    4.2    Optical modelling and measurements

To compare the $\beta_{NC}^{AR}$ backscattering calculated with optical modeling applied to the PSDs, we select point-by-point the AR that provides the best match with the $\beta$ measured by the backscattersonde. In order to illustrate the results, in the upper panels of Figure 3 we report the time series of the $\beta$ measured by the backscattersonde MAS during the flight on 10 Aug 2017 (red line), together with the best choice of the calculated $\beta_{NC}$ (black line), with representative error bars in the first part of the time serie.

The black dashed lines report the highest and lowest values of the $\beta_{NC}^{AR}$. The agreement between the measured backscattering $\beta$ and the selected one among the optical computations $\beta_{NC}$ can be made surprisingly good almost everywhere, except at its highest values - where the loss of linearity in the response of the backscattersonde could play a role - and at its lowest values - where it is possible that backscattering from particles below the minimum cloud spectrometer detectable size can become non-negligible. It is noticeable how in many parts of the time series the calculated value $\beta_{NC}$ is able to reproduce even the

finest structures of the measured $\beta$. From the figure we note that the selected $\beta_{NC}$ is often in the lower range of variability of the $\beta_{NC}^{AR}$ values. Moreover, the random error attributed to $\beta_{NC}$ is an order of magnitude lower than the uncertainty range deriving from the lack of knowledge in the particulate AR. In fact, the particle AR is probably the main factor of uncertainty to be attributed to our optical calculations.

As a test for the arbitrariness of the choice of AR that ensures the best agreement with the measurements, we looked for

possible correlations between the selected AR and i. the particle depolarization $\delta_A$; the fraction of aspheric particles AF. In the lower panels of Figure 3 the AR that provided the best match is reported (black line) together with the observed particle depolarization (red line) and particle AF, as measured by NIXE-CAPS (blue line). We note that the AR values that made the best match are arranged according to a certain temporal continuity, and we can identify well-defined time intervals in which their value remains almost constant. This is encouraging to think that the choice of AR that best matches the measures is not

exclusively an exercise of handpicking, but rather reflects characteristics of the average morphology of the measured particulate matter. It is of particular interest to note that regions with AR = 3 - that provides the lowest $\beta_{NC}^{AR}$ - almost always coincide with



regions in which the NIXE-CAPS has observed the highest particle AF. This, in turn, is often correlated with high values of total depolarization.

We have explored a possible lack of linearity in the response of the backscattersonde at high backscattered signals as a possible cause for the discrepancy between the highest calculated and measured $\beta$. The linear response of the backscattersonde was checked in the laboratory before the campaign deployment. This was done by screening the receiving optics with a series of calibrated gray filters and controlling the response of the instrument to pulsed light of various intensity. This procedure was repeated at various levels of background light. For the present study a further test was carried out by comparing the backscattering coefficient observed during the campaign by the backscattersonde in some of the thickest clouds, on 8 August 2017 between 05:17 and 05:50 UTC (19000s and 21000s) and on 10 August 2017 between 10:00 and 10:33 UTC (36000s to and 38000s), with what observed by the two MAL lidars on board the Geophysica. These two lidars point respectively upward and downward providing profiles of Backscatter Ratio and depolarization. We have used signals from the closest lidar range, in its partial overlap region, and processed them as if they came from backscattersondes. The average of the two lidar backscattering coefficients observed by them at a distance of 500 meters upward and downward was then compared with what observed by MAS. The data have been averaged over 60 s. The result is displayed in Figure 4. Despite the scattering of the data points, which might be attributed to a lack of vertical homogeneity of the cloud, it shows that there is a good correlation between the Backscattering coefficient measured in-situ and that measured at close ranges. The lidar backscatter signal is measured in photon counting mode. The linear dynamic rangeis identified to be below the count rate of approx. 1.5 MHz. The complete signal saturation is noticed at approx. 15 MHz count rate. For the cases reported here the signals are at the levels of some tens of KHz, i.e. well inside the linearity range. Thus we may exclude any loss of linearity. So we are tempted to exclude a severe loss of linearity of the backscattersonde and a consequent significant underestimation of the largest backscattering coefficients observed by it. Further sustaining our conclusion, we note that the overestimation of the optical model appears also in regions where saturation of the backscattersonde signal should not be expected, as instance, around 34000s on the 10 August 2017 flight.

Figures 5 and 6 report the two-dimensional histograms of occurrence of the measured-calculated backscattering coefficients and particle depolarization. While for the $\beta$ the agreement is good and the analysis on the whole campaign dataset comfirms what has already been shown in the previously discussed time series, i.e. a underestimation of the $\beta_{NC}$ for values of $\beta$ below $5 \ 10^{-5} \ \mathrm{km}^{-1} \ \mathrm{sr}^{-1}$, and a slight overestimation for high values in a relatively insignificant number of cases, the agreement for the depolarization is not good. From the inspection of Figure 6 we can see an attempt to achieve a 1-1 correlation between the two datasets. However, for a considerable number of observations, the calculated depolarization remains around 10% while the measured one varies throughout its range of variability. Moreover the optical modelling completely fails to reproduce the measured depolarization. We have tested how to improve the agreement by linking the choice of AR to the matching with depolarization rather than with backscattering. Of course, the comparison between measured and modeled depolarization improves but in the same way leads to a larger disagreement in the backscatter, which in many cases is strongly overestimated by the optical model by up to 1-2 orders of magnitude. (see figure A6 in Appendix). Even where the agreement between the calculated and the measured $\beta$ is kept within an order of magnitude, it fails to reproduce effectively the experimental dataset,





as reported in Figures 5 and 6. In some ways it is unfortunate not to be able to simulate backscattering and depolarization simultaneously with a single choice of AR. However, we are aware of the fact that the modeling of depolarization is an open and highly challenging problem, one that is not easy and perhaps impossible to solve given the wide variety of shapes that atmospheric ice crystals can take (Liou and Yang, 2016). For this reason we prefer to keep our comparison between optical measurements and calculations limited to the backscattering coefficient.

### 4.3  Backscattering versus Particle Size Distribution bulk parameters

Figures 7, 8, 9, 10 and 11 show $N_{ice}$, $R_{mean}$, $R_{eff}$, SAD and IWC as a function of $\beta$, respectively. In the graphs, the black lines represent regression curves, parametrized in the general form $X = A \cdot \beta^B$. The coefficients of the regression are reported in Table 1 together with the R-squared of the fit. The dataset has been fitted between the limits $2 \cdot 10^{-5}$ and $5 \cdot 10^{-3}$ km$^{-1}$ sr$^{-1}$, which have been chosen to maximize the goodness of the fit and try to avoid outliers at the extremes of the variability range. We can regard those regression curves as guidance for estimating the microphysics bulk parameters of the clouds and their variability range, from remote measurements of cloud optical parameters.

The linearity between the $\beta$ and $N_{ice}$ (Fig. 7) is quite striking and indicates that $\beta$ basically scales with the cloud particle number density $N_{ice}$, as seems to do SAD and IWC. Hence, this suggests that the various shapes of the PSD in our observations are hardly change the scattering properties of the clouds. This finding is further confirmed by the linearity seen between $\beta$ and SAD or IWC (Figs.10, 11), and the lack of clear correlation between $\beta$ and $R_{mean}$ or $R_{eff}$ (Figs. 8, 9).

This is most probably because the majority of measurements during StratoClim is at T < 200K, where there is only little water available thus in- situ origin ice particles cannot develop a variety in shapes – in fact most of them are so called "irregulars". Only the large liquid-origin ice particles can have distinct shapes, but they are rare.

### 4.4  Depolarization, Aspherical Fraction and PSD parameters

#### 4.4.1  Depolarization vs Aspherical Fraction

In our study we have looked for and found no direct correlation between particle depolarization observed with the backscattersonde, and PSD Aspherical Fraction (AF). AF has peaks around 60% but on the average is maintained around values of 20%, while the corresponding values of the depolarization spans its entire range of variability, giving an unclear relationship between the two quantities (see figure A8 in the Supplementary Material).

This is possibly because the maximum detectable AF of ice particles decreases with the size of the particles. The reason is that the depolarization signal becomes weaker with decreasing ice particle size and also the ice particles become lesser and lesser aspheric, and the backscattersonde and the cloud spectrometer have depolarization sensitivities that vary differently with the size of the scattering particle.

In order to display other parameters along with the former two, that may help to find and possibly disentangle a link between these two quantities, we choose to represent our dataset in a $\delta_T$ - BR space (in fact, for ease of scale, 1-1/BR is used instead of BR), and color code the points with AF, N, $R_{mean}$ and temperature T as possible parameters of interest.



We remind here that $\delta_T$ is not an intensive quantity as $\delta_{TA}$, since it simultaneously depends on the average shape of the

cloud particles and on the backscattering of the whole particle distribution, hence on the particle number concentration, or SAD, as instance. Hence, for a given and fixed particle shape and dimension, $\delta_T$ increases with the BR to a limiting value, which is $\delta_{TA}$. It is the latter the true intensive quantity that depends solely on the particle's average morphology. In fact, Adachi et al. (2001) demonstrated that in a plot of Total Volume Depolarization $\delta_T$ towards 1-1/BR, the experimental points of clouds composed of particles sharing the same shape and size but with variable particle number density (i.e. variable BR), will arrange

themselves along a straight line, starting at $\delta_{mol}$ for BR=1 (this is the case when no particles are present i.e. when the Total Volume Depolarization attains its molecular value $\delta_{mol}$ (Young, 1980)) and ending at a $\delta_{TA}$ for $BR = \infty$, this $\delta_{TA}$ depending on the particular shape and size common to all particles.

If we were to report data points from clouds composed of particles with different shapes and sizes, these points will be distributed within a triangle whose vertices are $\delta_{mol}$ for $BR = 1$, a $\delta_{TAmax}$ and $\delta_{TAmin}$ for $BR = \infty$. It should be noted that,

if among the possible particle shapes the spherical one is included, then $\delta_{TAmin}$ would attain a 0 value.

In Figure 12 we report our dataset, with the data points from in-cloud measurements (i.e. BR>1.2, 1-1/BR > 0.17) color coded in terms of AF. The y axis intercepts at $BR = \infty$ gives us the range of variability of $\delta_{TA}$, which for our cirrus ranges approximately from 20% to 50%.

We remind again that in this (BR, $\delta_T$) graph, ice clouds composed of particles with the same $\delta_{TA}$ and different BR (which we

may take as a proxy for particle number concentration) would be distributed along a line starting at $\delta_{mol}$ for $BR = 1$ and and trending towards a particular $\delta_{TA}$ when extrapolated to $BR = \infty$. The value of such $\delta_{TA}$ depends on the average morphology of the cloud particles. Therefore we may imagine the data point distribution as composed of linear series of points starting at $\delta_{mol}$ and spanning the triangle: each line of data points represents the results of the measured variable particle number concentration at constant depolarization properties of the particles as encountered in the clouds.

Inspecting Figure 12 we may argue that clouds with medium AF (cyan colour) are often linked to intermediate values of $\delta_{TA}$ over the whole BR variability range, while clouds with high AF (yellow and red colour) , associated with high BR, show no clear correlation with $\delta_{TA}$. Interestingly, clouds with low AF (blue colors) show up at both low BR (with low to intermediate depolarization) and high BR (with intermediate to high depolarization). The conditions at the highest BRs are in fact noteworthy: there, we have both the highest values of the AF for $\delta_{TAmin}$ and $\delta_{TAmax}$, and the lowest values of the AF in

a range spanning from $\delta_{TAmed}$ to $\delta_{TAmax}$.

These relationships between BR, depolarization and AF are probably reflecting aspects of the morphology, size and numerical concentration of the cloud particulate, but are not straightforward to interpret unambiguously. To seek a better understanding, we present in similar plots the dependence of depolarization on $N_{ice}$ and $R_{mean}$ , and on temperature.

### 4.4.2 Depolarization vs Particle Number Concentration, mean Radius, Temperature

We report in Figure 13, using the same $\delta_T$ - BR framework, the data points color coded in terms of $N_{ice}$ and in Figure 14 the same data points, color coded in terms of $R_{mean}$ . Figure 13 clearly shows again the positive correlation between BR and $N_{ice}$, independent of the polarization $\delta_{TA}$ except at the highest values of BR, where it appears a positive correlation





between $\delta_{TA}$ and $N_{ice}$. Figure 14 shows how high $\delta_{TA}$ are often associated with high $R_{mean}$ (red dots along the line $\delta_{mol}$-$\delta_{TAmax}$), and again peculiarly in the same region of highest values of BR, small $R_{mean}$ coincides with medium to high $\delta_{TA}$).

In Figure 15 we show the same dataset this time color coded in terms of temperature. The points at temperatures above 200K are showing both very high and, although in smaller numbers, very low depolarization with intermediate to high BR values. Points below 200K show the general tendency to an increase in depolarization at colder temperature, already commented in figure 2. Noteworthy, the observations in the region of very high BR and between $\delta_{TAmed}$ and $\delta_{TAmax}$, where the coldest temperatures, large particle concentration, low AF and low mean radius have been met, are worth of a special mention. These

all came from clouds encountered in a single flight, performed on 10 August 2017. On that flight, a convective overshooting updraft of probably liquid origin, with very large aggregates and significant number of freshly nucleated cloud particles was met, together with younger outflow of the overshoot with both growing small particles and sedimenting large ones ((Krämer et al., 2020) see their Figure 10b, (Khaykin et al., 2022). This dynamic and therefore particularly variable situation makes the overall interpretation of the observed parameters exceptionally complicated.

As stated at the beginning of the paragraph, there seems to be no direct correlation between $\delta_{TA}$ and AF and although some clustering of the variables is possible, these clusters often overlaps. It is possible that additional information on particle shape in this analysis, which is missing in the present study (a measured aspect ratio would be an obvious candidate), would help disentangle the variables.

## 5 Discussion

As outlined in the previous paragraph, in our study the T-matrix has been used directly to model the scattering properties of particles up to a few hundred size parameters, and indirectly to estimate the depolarization, and the modification of Mie backscattering, for larger particles. We remind that while Mie theory strictly applies to spherical particles, it has no upper size limit and can be applied to the entire cloud particle size range.

Despite the limitations outlined above, some conclusion can be deduced from our study. It is possible to make the measure-

ments coincide with the optical modeling through a suitable and reasonable choice of the particle AR, and this reassures us about the compatibility of the two datasets. This choice generally favors high ARs (i.e high Mie backscattering depression) in regions of high backscattering, high depolarization and large presence of AF, while ARs near or less than unity are chosen when backscattering is medium to low and the presence of AF is not very pronounced. In general the choice of AR produces a high depression of the backscattering compared to Mie's predictions, and this depression increases with increasing presence of

aspherical scatterers, reaching to exceed an order of magnitude when AF is large. In cases with particularly high backscattering, our optical model produces backscattering higher than those observed. If - as seems appropriate - non-linearity problems in the backscatter probe are excluded, it is quite possible that, to reduce the computed backscattering and reconcile it with observations, we would need to use ARs beyond the limits considered in our optical model.

Moreover, it is clear from our simulations that the greatest ambiguity in the results of the comparison is linked to the choice

of particles' morphology, rather than to uncertainties in the determination of the PSD or in the measurement of backscattering.





Another interesting result is the identification of the size range of the particulate matter that most contributes to the observed backscattering. For all the PSD under study we have examined the cumulative function (19)

$$\beta_{NC}^{Mie}(r_i) = \frac{\sum_{i=1}^{r_i} \beta_i^{AR}}{\beta_{NC}^{Mie}} \qquad (19)$$

of the respective $\beta$ as a function of the particle radius, in order to determine the buildup of the final values of the particle

backscattering coefficient in relationship to the particle radius. We have used Mie codes to give an upper limit to such computation. The histogram of the values of the cumulative distributions as a function of the particle radius is reported in Figure 16 . It can be seen, from the analysis of the histogram of cumulatives, that the values of the backscattering coefficient are formed mainly in the dimensional range below 300 $\mu$m and particles with radii greater than 400 $\mu$m do not significantly contribute to the $\beta$. This analysis gives us confidence in affirming that the size range of the detected particles is sufficient to fully characterize

the backscattering coefficient.

The T-matrix computation of depolarization gives no satisfactory results, being the modelled depolarization an underestimation in comparison to the depolarization measured by MAS,. It is possible that this is due to the specificity of our approach. Our computations are not able to produce $\delta_A$ higher than 40% in the large particle regime (i.e. radius grater than 10 $\mu$m). Other theoretical depolarization ratios computed from a unified theory of light scattering by ice crystals for shapes including bullet

rosettes, solid and hollow columns, Koch snowflakes, and plates, extend up to 60% and more (Liu and Mishchenko, 2001). As stated previously there seems to be no clear correlation between particle depolarization and AF. Particle depolarization increases with temperature, with N$_{ice}$ at high backscattering. High depolarization is often, but not always, associated with high R$_{mean}$ as there are cases when at high values of backscattering, small R$_{mean}$ coincides with medium to high $\delta_{TA}$. This study therefore suggest that, although some correlations can be discerned between the depolarization and the environmental condi-

tions in which the cloud is observed, or its microphysical and morphological average parameters, there are a lot of exceptions. These probably depend on the history of the formation and the instant of evolution of the cloud, leading to the coexistence of cloud particles with different morphology and sizes. Hence, the study does not allow us to formulate general rules that link depolarization to the microphysics of the cloud.

The regression of the backscattering coefficient with respect to the various bulk parameters of the cirrus clouds $N_{ice}$, $R_{mean}$,

$R_{eff}$, SAD, IWC therefore is based on a solid foundation. Effectively, similar regressions were presented in Cairo et al. (2011) based on cirrus particle measurements from an FSSP-100. There, the upper detection limit of the instrument was 15.5 $\mu$m in radius, while in the present study this limit has extended upward for more than an order of magnitude. So, although the present figures are qualitatively similar to those of Cairo et al. (2011), some difference is to be expected due to the larger range of particle radii that is now covered. The limitedness of the radius range was a caveat already advocated in the aforementioned

work. In fact the larger detection threshold is impacting all the regressions, which results in a underestimation in Cairo et al. (2011) of $N_{ice}$, SAD and VD. In the present work, the detection of a wider particle dimensional range, including the radii which determine most of the measured backscatter coefficient, allows us to place more reliability on the regressions here presented.

The relative independence of $\beta$ from $R_{mean}$ and $R_{eff}$ confirms $N_{ice}$ as the main parameter governing the cirrus scattering properties at optical wavelengths. This finding would imply that the shapes of the PSD should not play a mayor role and should



all share a similar shape once normalized for the total number of particles, at least for low to medium $\beta$ values. For high $\beta$ values, the spreading of the corresponding PSD get larger, suggesting that such observations originated from both clouds of very large and few particles, and of small and more numerous particles. Konoshonkin et al. (2017b) computed backscattering Mueller matrix for the typical shapes of ice crystals of cirrus (hexagonal columns and plates, bullets and droxtals) in the case of their random orientations, for crystal size from 10 $\mu$m to 1000 $\mu$m with a physical-optics-approximation code and

proposed to use the backscatter-to-IWC ratio (as well as extinction-to backscatter, (LR) ) for inferring the crystal size in the clouds. Figure 17 reports such ratio versus $R_{mean}$ which in fact shows such linear relationship for $R_{mean}$ larger than 10 $\mu$m, while for lower values a different linear behavior might be discerned. In the Figure, the black line is a regression of the form $\frac{beta}{IWC} = 1.52 \cdot 10^{-3} \cdot R_{mean}^{-2.34}$ which has been estimated for $R_{mean}$ from 10 to 100 $\mu$m. The R-squared value of the fit is 0.54, which does not allow us to conclude that such possible linear relationship is based on solid foundations. Because of this, such

regression should be used with great caution to estimate $R_{mean}$ when $\beta$ and IWC are independently available.

Estimates of IWC can be obtained from lidar extinction $\sigma$ measurements (Heymsfield et al., 2005). Such estimates assume a relationship of the form IWC=A $\sigma^B$ between the two values, where A and B which in Heymsfield et al. (2005) were posed to A=119 and B=1.22. In a later study Heymsfield et al. (2014) these coefficients have been found to depend on the temperature, based on in situ measurement of IWC and particle PSD obtained during the field campaign of the Stratospheric–Climate

Links with Emphasis on the Upper Troposphere and Lower Stratosphere (SCOUT) project, based out of Darwin, Australia, in November–December 2005 (de Reus et al., 2009).

The IWC estimates from the knowledge of the extinction $\sigma$ could be compared with our IWC estimates based on $\beta$, if a suitable LR is chosen. Unfortunately, LR can vary from 10 to 40 sr in tropical cirrus clouds (Chen et al., 2002) thus making the comparison somewhat arbitrary. Using a LR = 30 sr and posing $\sigma = LR \cdot \beta$ we correlated the IWC measurements with

their estimate obtained with the regression towards $\beta$ presented in this work, and with one obtained from the formulation of Heymsfield et al. (2014) . Although both are successful in predicting the IWC within the order of magnitude, the estimate obtained with regression using directly $\beta$ yelds better results. This is probably due to having chosen the same LR value for all the clouds observed. It should also be noted that the $\sigma$-based estimation improves by using the formulation with coefficients that do not depend on the temperature, as in Heymsfield et al. (2005), see figure A9 in Appendix.

To conclude, a comment on the representativeness of our study. The range of backscattering values observed during campaign activities is wide and covers the variability of possible lidar observations from satellite (Balmes et al., 2019). Regarding the type of cirrus clouds observed, during the campaign activities both in-situ and liquid-origin were sampled, but the second type is dominating since most of the observations came from penetration into the outflow regions of deep convective clouds. As there might be differences in the microphysical properties of the cirrus depending on the formation process (Krämer et al.,

2020) especially in the initial stage of their life cycle, this may induce an unquantified bias in our presented statistics.



## 6 Conclusions

Measurements in cirrus clouds obtained during the StratoClim campaign by two dual-instrument cloud spectrometers, two hygrometers and a backscattersonde have been compared. The comparison of the microphysical data with the optical observations of the backscattersonde was performed by calculating the ice particle backscattering coefficient from the PSD by means of

optical modelling. A proper adjustment of the modeled particle AR allows to match the optical computation with the backscattering observations. Relations have been obtained to link the ice particles backscatter coefficient to their concentration, their mean and effective radius, surface area density and ice water content. These results confirm and expand similar studies and allow to estimate within an order of magnitude, data of bulk cirrus microphysics from lidar remote sensing observations.

The comparison between particle depolarization from the backscattersonde and the Aspherical Fraction measured by one of

the cloud spectrometers shows no univocal relationship between the two quantities and ask for further investigations, where additional information on particle morphology may be required.

*Code and data availability.* Water measurements (FLASH and FISH), meteorological data from UCSE and TDC, MAS backscatter and

particle measurements from NIXE-CAPS from the StratoClim aircraft campaign are available on the HALO database at https://halo-db.pa.op.dlr.de/mission/101 (DLR, 2022). Particle measurements from CCP are available upon request to the PIs. Lidar measurement from MAL are available upon request to the PI. Analysis and plotting scripts for this paper are available upon request to the corresponding author.

*Author contributions.* MS, LDL, FC prepared the MAS for campaign activity. FC performed and processed the MAS measurements. LL

did the analysis scripts for data comparison. SK performed and processed the FLASH measurements. CR, AA, NS and MK performed and processed the FISH and NIXE-CAPS measurements. SB, MP performed and processed the CCP measurements. RW contributed to the




collection of the particle dataset. SB and FC conceived and planned the study. FC performed the analysis, prepared the figures, and drafted the manuscript. All authors commented on the manuscript.

*Competing interests.*  The contact author has declared that none of the authors has any competing interests.

*Acknowledgements.*  We gratefully thank the StratoClim coordination team and the Myasishchev Design Bureau for successfully conducting the field campaign. This research has been supported by the StratoClim project of the European Community's Seventh Framework Programme (FP7/2007–2013) under grant agreement no. 603557 and internal funds of the Max Planck institute for Chemistry. FC is grateful to
the MPI for Chemistry for supporting his sabbatical stay in Mainz during which the study behind this publication was initiated. SK's work was partly supported by the Agence Nationale de la Recherche TTL-Xing ANR-17-CE01-0015 project.

## Appendix A:  SUPPLEMENTARY MATERIAL

We include in this Appendix in figure A1 and A2 the particle backscattering coefficient and depolarization vs particle radius, parametrized with the particle AR. These optical parameters have been computed for a reference monodisperse PSD with
particle concentration of 1 $cm^{-3}$. In figure A3 the 2D histogram of the particle concentration $N_{ice}$ computed from PSD from NIXE-CAPS (x-axis) and CCP (y-axis). A4 reports the 2D histogram of the particle mean radius $R_{mean}$ computed from PSD from NIXE-CAPS (x-axis) and CCP (y-axis). In both figure, for NIXE-CAPS only particles with with lower size limit at 1.5 $\mu$m have been considered. A5 the comparison of the backscattering coefficients retrieved from the PSD measured by the two cloud spectrometers. In A6 results of the backscattering comparison in the case when the AR are chosen to make the depolarization
match. In figure, time series of particle backscattering coefficient $\beta$ measured by MAS on 10 August 2017 (red lines); Black solid line, $\beta_{NC}$ corresponding to the best matching between measured $\delta$ and computed $\delta_{NC}^{AR}$ (this latter in not displayed); error bars are reported in the first part of the curve; black dashed lines, maximum and minimum values of the optical modeling $\beta_{NC}^{AR}$ values. In the lower panel: Red line , particle depolarization measured by MAS; Blue line Particle Aspherical Faction measured by NIXE-CAPS; black line, AF values of the best matching between measured $\delta$ and computed $\delta_{NC}^{AR}$.
In figure A7 we report 2D histogram of the backscattering coefficient computed with PSD with lover size limit at 0.3 $\mu$m (y-axis) and at 1.5 $\mu$m (x-axis).

In figure A8 we report 2D histogram of Particle AF (y-axis) vs Particle depolarization (x-axis).

Finally, in figure A9 scatterplots of estimated vs measured IWC using different regressions are reported. In the left panel the regression from the present work is presented. This uses the measured $\beta$ to estimate the IWC. In the middle and left panels,
the regressions from Heymsfield et al. (2014) (middle panel) and from Heymsfield et al. (2005) (right panel) are reported.





These two have been obtained by using the measured $\beta$ to estimate the extinction $\sigma$, originally used in Heymsfield's works, and assuming a LR=30 sr. The colors code the temperature at the observation.



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





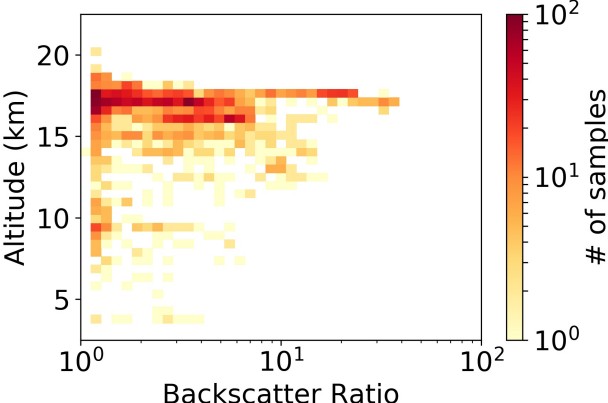

**Figure 1.** 2D histogram of Backscatter Ratio vs altitude. Data were acquired throughout the campaign by the MAS backscattersonde. The color codes the number of observations in the 2D bin.





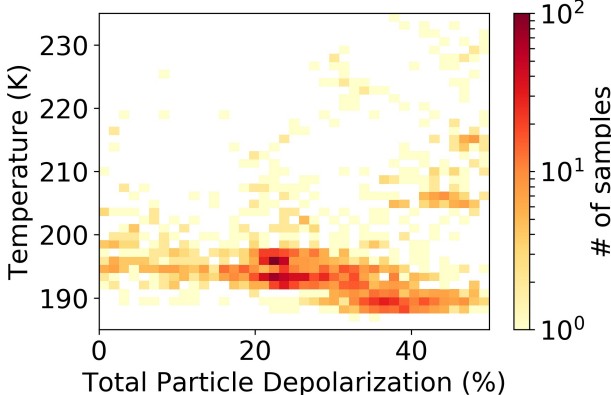

**Figure 2.** 2D histogram of Total Particle Depolarization data vs temperature for altitudes above 11 km. Data were acquired throughout the campaign by the MAS backscattersonde. The color codes the number of observations in the 2D bin.

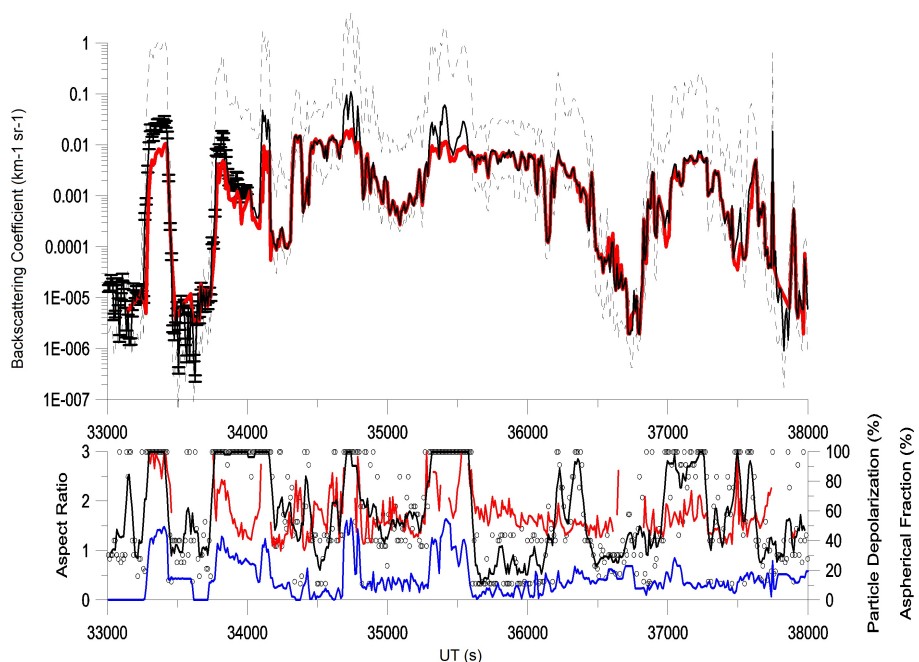

**Figure 3.** Upper panel: Red line, time series of particle backscattering coefficient $\beta$ measured by MAS on 10 August 2017; Black solid line, best optical modelling match $\beta_{NC}$, error bars are displayed in the first part of the curve; black dashed lines, maximum and minimum values of the optical modelling $\beta_{NC}^{AR}$ values. Lower panel: Red line , particle depolarization measured by MAS; Blue line Particle aspherical fraction measured by NIXE-CAPS; black dots, AF values of the best optical modeling match $\beta_{NC}$, the black line is a 5-points running average through them.



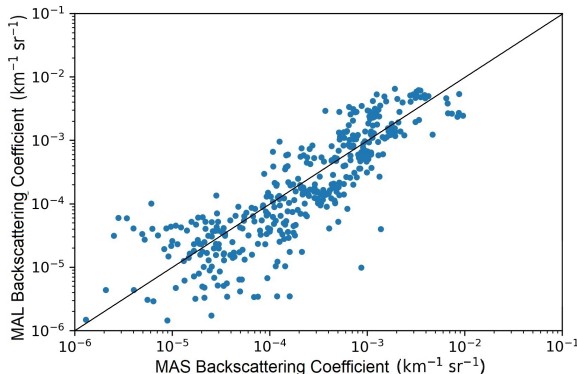

**Figure 4.** 2D histogram of particle backscatter coefficient observations from the backscattersonde MAS and the particle backscattering coefficients data from the two lidars MAL, pointing upward and downward, at 500m from the aircraft. The average of the two lidar backscattering coefficients has been used for the comparison. The data have been averaged over 60 s. In the graph are reported 500 data points acquired while crossing some of the thickest clouds observed during the campaign.



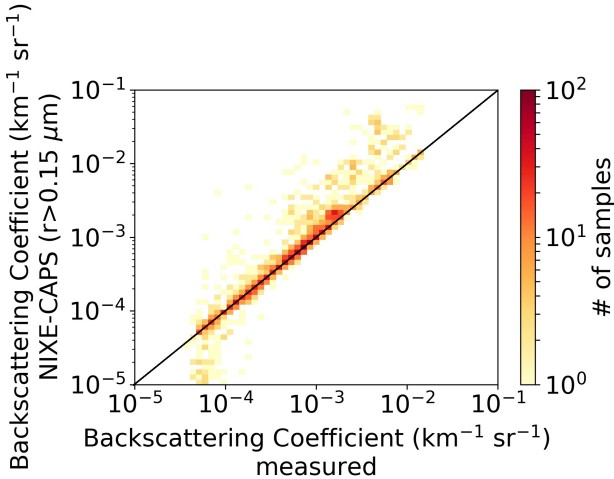

**Figure 5.** 2D histograms of occurrence of the measured and calculated backscattering coefficients. Data were acquired throughout the campaign by the MAS backscattersonde. The color codes the number of observations in the 2D bin.



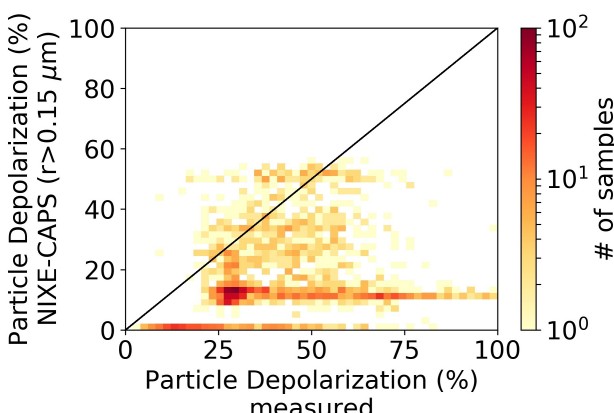

**Figure 6.** 2D histograms of occurrence of the measured and calculated particle depolarization. Data were acquired throughout the campaign by the MAS backscattersonde. The color codes the number of observations in the 2D bin.



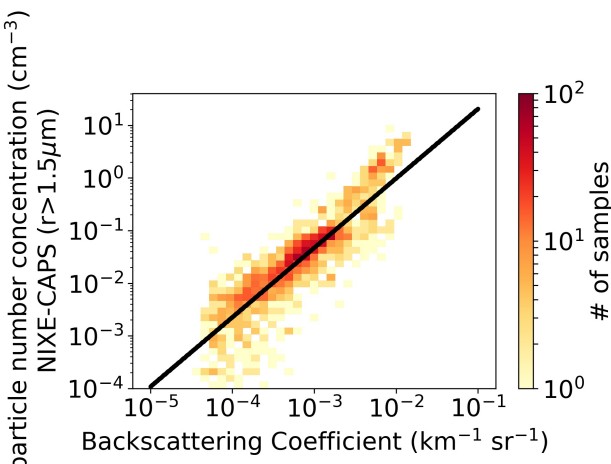

**Figure 7.** 2D histogram of particle backscattering coefficient observations $\beta$ from the backscattersonde MAS and particle number concentration N. The black line represent the fit N=4.33·$10^2$ ·$\beta^{1.32}$. Data were acquired throughout the campaign by the MAS backscattersonde. The color codes the number of observations in the 2D bin.



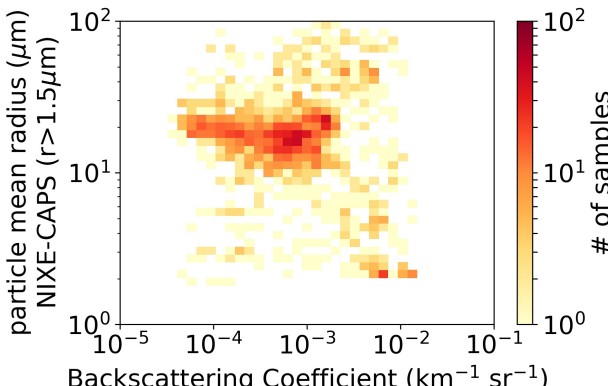

**Figure 8.** 2D histogram of particle backscattering coefficient observations $\beta$ from the backscattersonde MAS and particle mean radius $R_{mean}$. Data were acquired throughout the campaign by the MAS backscattersonde. The color codes the number of observations in the 2D bin.



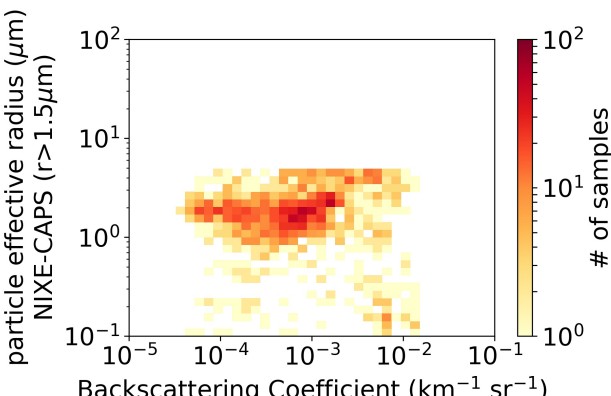

**Figure 9.** 2D histogram of particle backscattering coefficient observations $\beta$ from the backscattersonde MAS and particle effective radius $R_{eff}$. Data were acquired throughout the campaign by the MAS backscattersonde. The color codes the number of observations in the 2D bin.





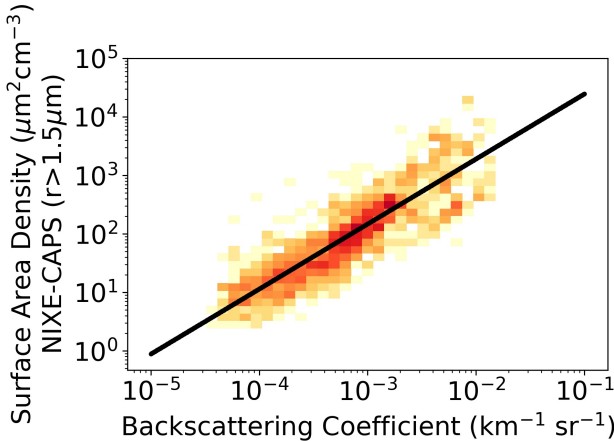

**Figure 10.** 2D histogram of particle backscattering coefficient observations $\beta$ from the backscattersonde MAS and particle Surface Area Density SAD. The black line represents the fit SAD=$3.24 \cdot 10^5 \cdot \beta^{1.11}$. The color codes the number of observations in the 2D bin.



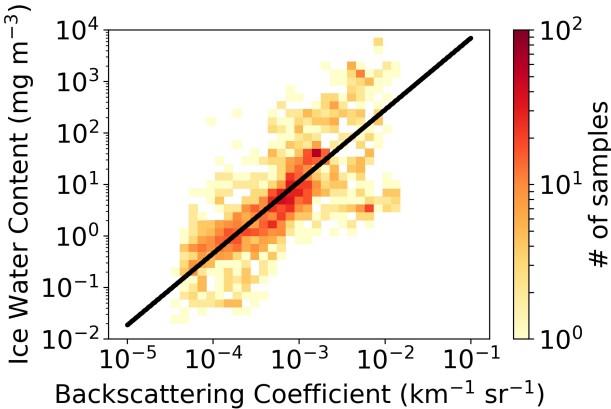

**Figure 11.** 2D histogram of particle backscattering coefficient observations from the backscattersonde MAS and Ice Water Content IWC. The black line represents the fit IWC=$1.17 \cdot 10^4 \cdot \beta^{1.39}$. The color codes the number of observations in the 2D bin.



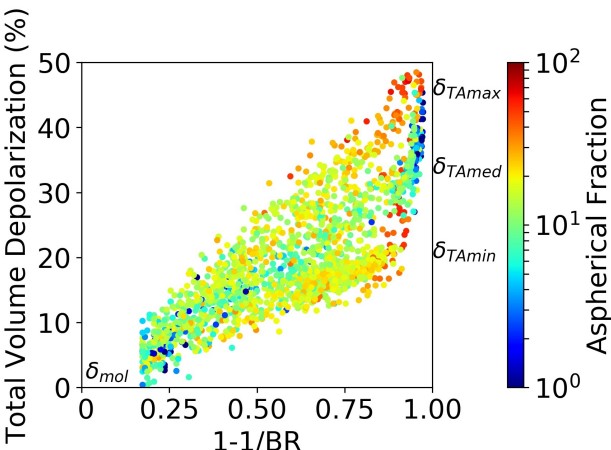

**Figure 12.** Scatterplot of BR vs $\delta_T$ color coded in term of particle Aspherical Fraction.



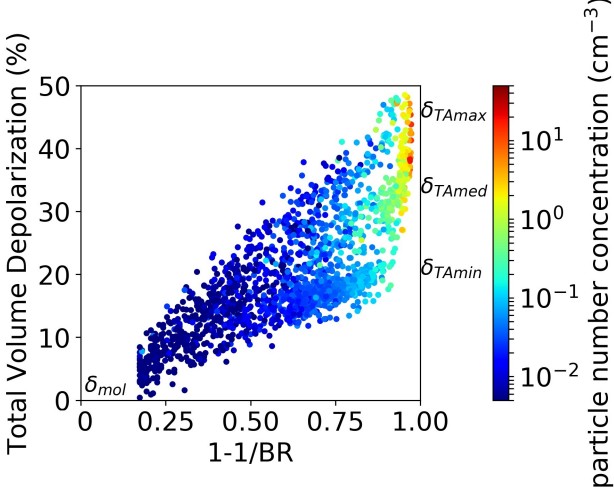

**Figure 13.** Scatterplot of BR vs $\delta_T$ color coded in term of particle number concentration.



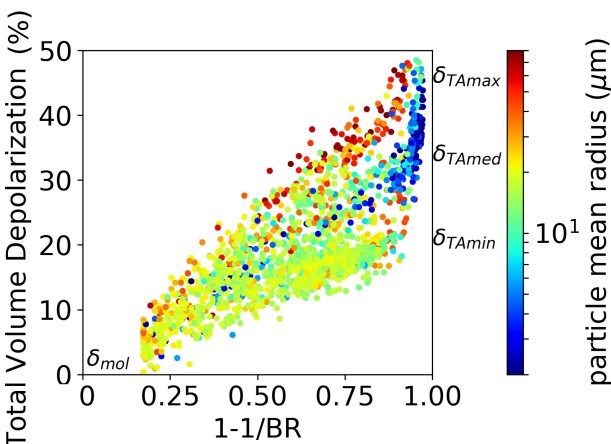

**Figure 14.** Scatterplot of BR vs $\delta_T$ color coded in term of particle mean radius.




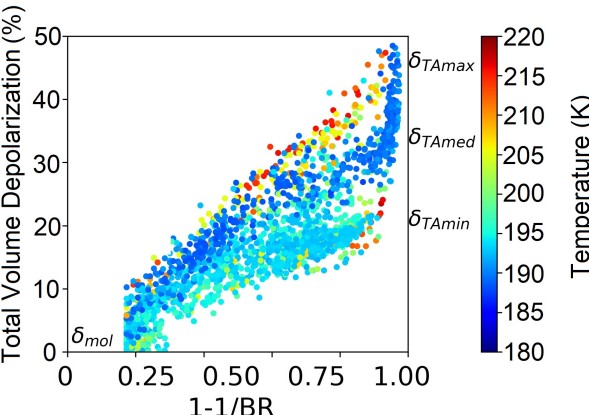

**Figure 15.** Scatterplot of BR vs $\delta_T$ color coded in term of temperature.

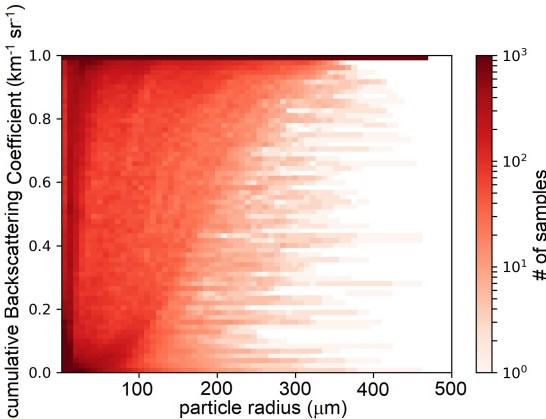

**Figure 16.** 2D histogram of cumulative distributions of particle backscattering coefficient computation. The histogram shows the buildup of the particle backscattering coefficient with respect to the particle radius, for all the PSD under study. For the present graph, Mie scattering computation have been used.





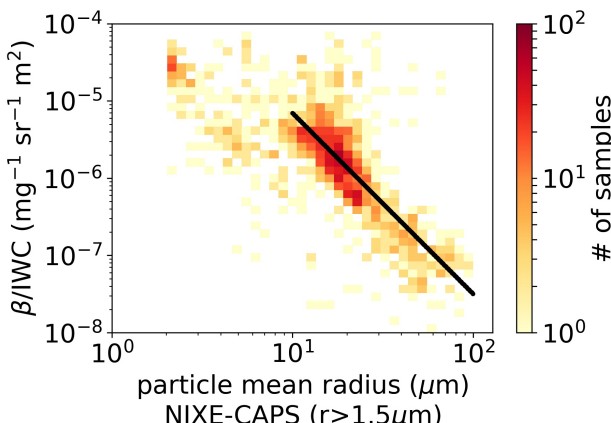

**Figure 17.** 2D histogram of the ratio of the backscattering coefficient to IWC, versus particle mean radius. The black line report a regression computed for $R_{mean}$ from 10 to 100 $\mu$m. The fit is $\frac{beta}{IWC} = 1.52 \cdot 10^{-3} \cdot R_{mean}^{-2.34}$. The color codes the number of observations in the 2D bin.

**Table 1.** Linear relations linking the backscattering coefficient to particle number, surface (SAD) and Ice Water Content (IWC). Here $\beta$ is expressed in km$^{-1}$ sr$^{-1}$ while N, SAD and IWC are expressed respectively in cm$^{-3}$, $\mu$m$^2$cm$^{-3}$, and mg m$^{-3}$.

| fit | R-squared |
|---|---|
| N=4.33$\cdot$10$^2$ $\cdot\beta^{1.32}$ | 0.71 |
| SAD=3.24$\cdot$10$^5$ $\cdot\beta^{1.11}$ | 0.73 |
| IWC=1.17$\cdot$10$^4$ $\cdot\beta^{1.39}$ | 0.59 |





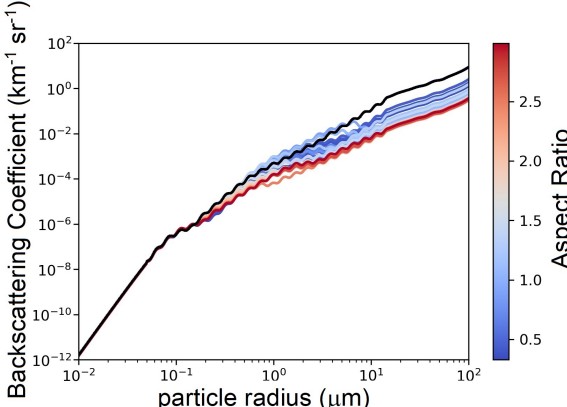

**Figure A1.** Particle backscattering coefficient vs particle radius, for various choices of the Aspect Ratio. A particle concentration of 1 $cm^{-3}$ has been considered. The black line refers to Mie computations (AR=1). For particle radius below 14 $\mu$m, the computations were performed with the GRASP package. Beyond 14 $\mu$m, for every given AR, the backscattering coefficients have been computed from Mie backscattering efficiencies, suitably rescaled with a constant factor. This scaling factor was chosen to make the scaled Mie efficiency to overlap with the GRASP T-matrix efficiency in the particle radius dimensional region from 5 to 14 $\mu$m (60-180 size parameters).

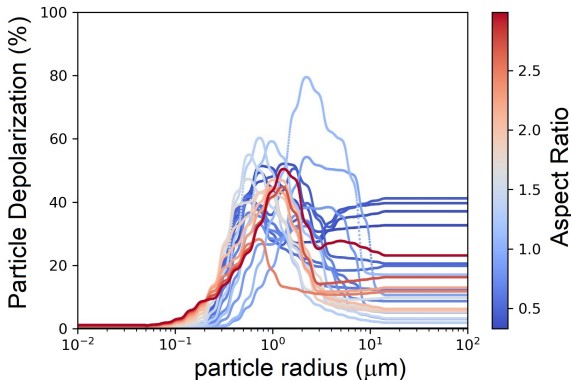

**Figure A2.** Particle depolarization vs particle radius, for various choices of the Aspect Ratio. The black zero depolarization line refers to Mie computations (AR=1). For particle radius below 14 $\mu$m, the computations were performed with the GRASP package. Beyond 14 $\mu$m, for every given AR, the depolarization of the particles was set at its constant, asymptotic value, computed as its mean over the the particle radius dimensional region from 5 to 14 $\mu$m (60-180 size parameters)



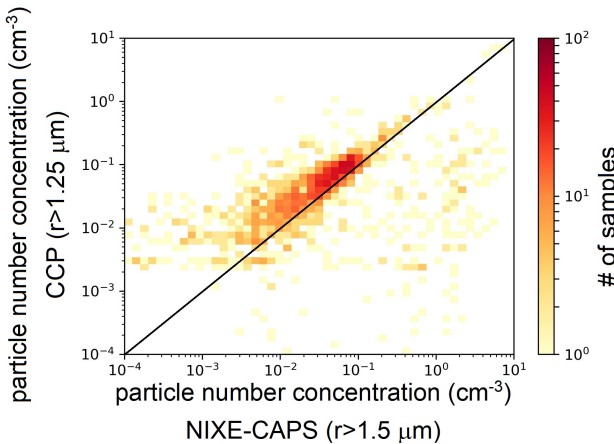

**Figure A3.** 2D histogram of the particle concentration $N_{ice}$ computed from PSD from NIXE-CAPS (x-axis) and CCP (y-axis). For NIXE-CAPS only particles with with lower size limit at 1.5 $\mu$m have been considered.

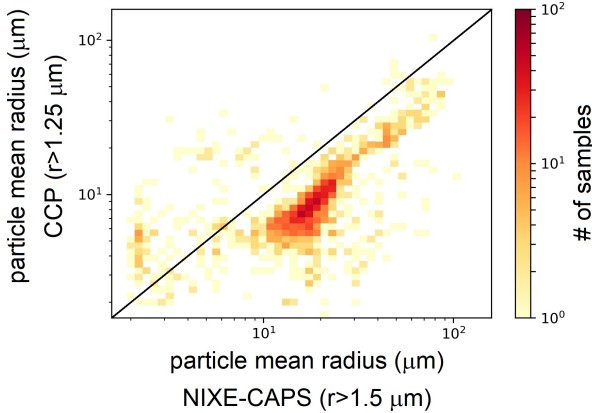

**Figure A4.** 2D histogram of the particle mean radius $R_{mean}$ computed from PSD from NIXE-CAPS (x-axis) and CCP (y-axis). For NIXE-CAPS only particles with with lower size limit at 1.5 $\mu$m have been considered.





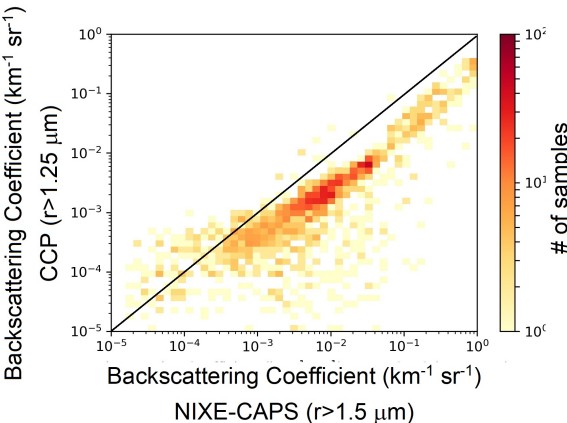

**Figure A5.** 2D histogram of the backscattering coefficient computed from PSD from NIXE-CAPS (x-axis) and CCP (y-axis). For NIXE-CAPS only particles with with lower size limit at 1.5 $\mu$m have been considered. Computation has been performed with Mie scattering codes.

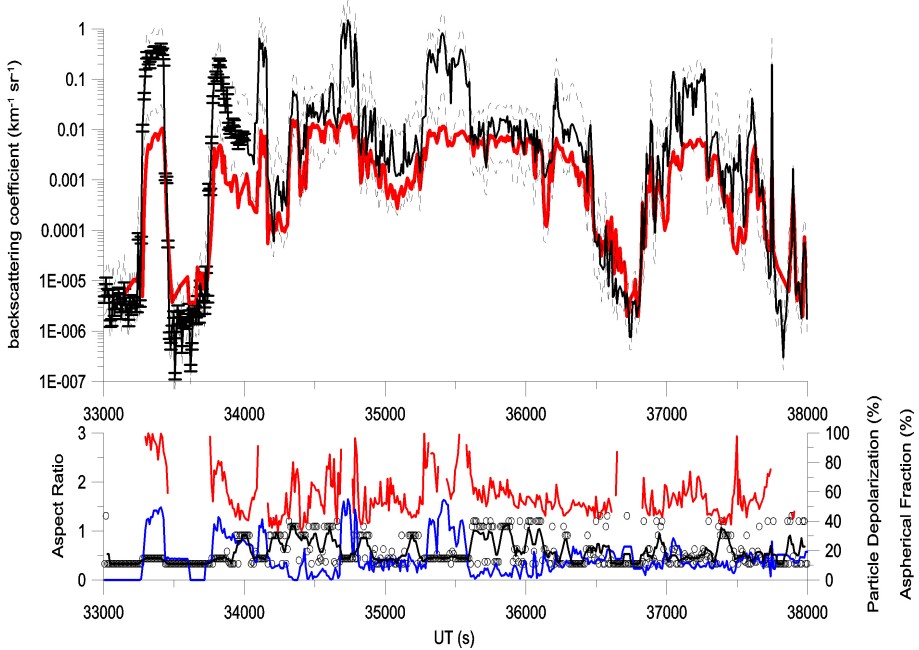

**Figure A6.** Red line, time serie of particle backscattering coefficient $\beta$ measured by MAS on 10 August 2017; Black solid line, $\beta_{NC}$ corresponding to the best matching between measured $\delta$ and computed $\delta_{NC}^{AR}$ (not displayed), error bars are reported in the first part of the curve; black dashed lines, maximum and minimum values of the optical modeling $\beta_{NC}^{AR}$ values. Lower panel: Red line , particle depolarization measured by MAS; Blue line Particle aspherical fraction measured by NIXE-CAPS; black line, AF values of the best matching between measured $\delta$ and computed $\delta_{NC}^{AR}$.





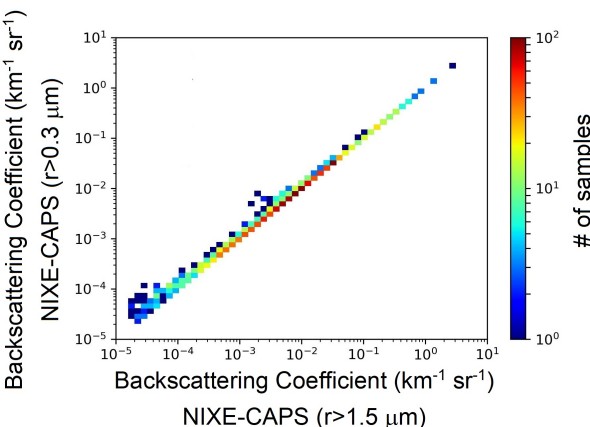

**Figure A7.** 2D histogram of the backscattering coefficient computed with PSD with lower size limit at 0.3 $\mu$m (y-axis) and at 1.5 $\mu$m (x-axis)

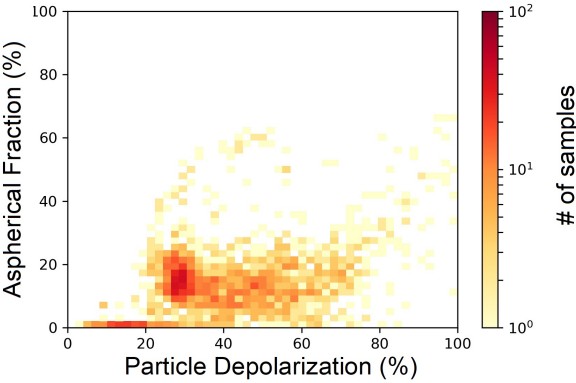

**Figure A8.** 2D histogram of Particle Aspherical Fraction (y-axis) vs Particle depolarization (x-axis)

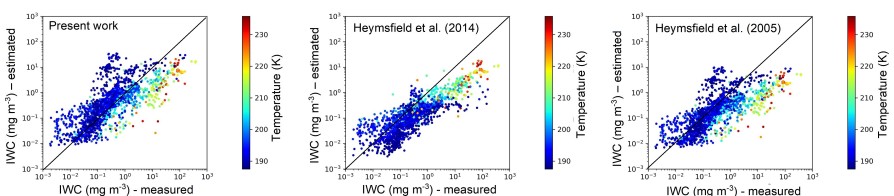

**Figure A9.** Scatterplot of estimated vs measured IWC using regressions from the present work (left panel), from Heymsfield et al. (2014) (middle panel) and from Heymsfield et al. (2005) (right panel). The colors code the temperature at the observation.