# Peer review of "A Comparative Analysis of In-Situ Measurements of High Altitude Cirrus in the Tropics"

_EGUsphere, 2023_

## Referee Comment (RC1)

Review of "*A Comparative Analysis of In-Situ Measurements of High Altitude Cirrus in the Tropics*," by Cairo et al., MS No.: egusphere-2023-112.

This is an interesting article that uses the M55 Geophysica aircraft, to relate in-situ measurements of particle size distributions and the individual particle properties, to spectrographic and lidar measurements. The measurements were conducted in high altitude cirrus clouds, near the tropopause. Particle probe instrument comparisons are, as well as comparisons between spectrographic and lidar measurements, are conducted. Relationships are derived between the bulk particle properties-effective radius, ice water content, etc, and their backscatter properties.

I have numerous comments that should be considered in the revision of the manuscript. My major and minor comments appear below.

**Major Comments**

1. The discussion of instruments covers pages 5-8, and the analysis methods between 9 and 12, which I think is excessive, given that the written text is 20 pages in total. And there is virtually no discussion of the clouds sampled-the article is more of an instrument comparison study. There's a little discussion (a few sentences) just before the conclusions section but should have much more detail earlier in the manuscript. A discussion of earlier measurements in similar types of cloud conditions is warranted.

2. There are numerous acronyms used throughout the article. A table of the names associated with acronyms is warranted.

3. Another point: Given that your PSD instruments only measured up to 960 microns yet you are sampling convective outflow cirrus, is this a problem? Do you have any particles that are in the largest size bin?

4. Lines 398-400. It's a very smart idea to compare backscattersonde to the lidar data

5. Figure 4. Can you change the units to something that might be more meteorologically oriented, for example $m^{-1}$? This is very much the case for Figure 7-11, and ones that follow.

6. Section 4.3, Figures 7-1q. I feel strongly that the units for backscattering coefficient in the figures units $km^{-1}$ $sr^{-1}$ should be something that modelers, etc could use. These should be put in standard cloud physics units. Also, IWC should be in $g/m^3$. This would facilitate comparison with other studies (for example, IWC in Figure 11 to Thornberry *Thornberry, T. D., A. W. Rollins, M. A. Avery, S. Woods, R. P. Lawson, T. V. Bui, and R.-S. Gao (2017), Ice water content-extinction relationships and effective diameter for TTL cirrus derived from in situ measurements during ATTREX 2014, J. Geophys. Res. Atmos., 122, 4494–4507, doi:10.1002/.*

7. Lines 524-525. The greatest ambiguity in the results of the comparison is linked to the choice of particles' morphology. Perhaps you should temper this because the mass dimensional relationship and the masses of the small particles are not compared directly to FLASH.

8. The relative independence of β from Rmean and Reff confirms Nice as the main parameter governing the cirrus scattering properties at optical wavelengths. Does this result also fall out of the analytical relationships assuming gamma PSD and quasi-spherical ice particles?

Minor Comments

I have numerous minor comments that should be considered in the revision of the manuscript.

Line 3. in view to > with the goal of connecting

7. Hymalaian to Himalayan

14? What do you mean by "can be set"

26: Cirrus at higher altitudes. Regarding your statements about cirrus, it would be good to use the AMS Glossary of Meteorology definition.

58. Why cite only recent studies. You could add **Heymsfield**, A. J., and R. G. **Knollenberg**, **1972**: Properties of cirrus generating cells. J. Atmos. Sci., 29, 1358–1366, for example

85. properties. This is particularly…

98. Why not use the Self-Similar Rayleigh-Gans Approximation (Hogan and Westbrook, 2014), or DDSCAT?

118. developed  by

265. This line should be part of paragraph on line 264.

285-330 Very nice, comprehensive calculations of how the aspect ratio affects the backscattering efficiencies.

339. Could you include the m-D relation you use in the text?

368-370 Very good determination of why the NIXE-CAPS data set was used.

Section 4.2. It's clever to use the backscatter model together with the measurements for find the best AR.

385. Remove i

405. backscattering should not be capitalized

408. range is

440. as seems to do > as well as SAD and IWC

441. as hardly change. Please rephrase

475 and and

512. We remind. We note that...

---

## Author Response (AR1)

**Point by point changes in the manuscript**

**For the Editor:** Please note I have added an author (Guido Di Donfrancesco) and updated one figure with corrected data (Figure 11).

- The discussion of instruments and the analysis methods have been reduced.
-  A discussion of earlier measurements in similar types of cloud conditions has been introduced.
- A table of the names associated with acronyms has been introduced.
- Backscatter coefficient now is in m-1. IWC now is in g m-3
- Figure A9 has been suppressed while a new comparison figure (Fig. 18) has been inserted in the text, with several IWC-▯ regressions, illustrated in a new table (Table 2).
- A table summarizing instruments has been added (Table 1).
- Figure 1 has been updated as requested.
- A new figure (Fig. A7) has been added to the Supplementary Material and commented in the text
- A new figure (Fig. A6) has been added to the Supplementary Material and commented in the text

**Answers to the First Reviewer's Review of "A Comparative Analysis of In-Situ Measurements of High Altitude Cirrus in the Tropics", by Cairo et al., MS No.: egusphere-2023-112.**

The Authors thank the Reviewer for the careful analysis of the manuscript, which led to a reconsideration and expansion of sections not sufficiently clear and/or exhaustive. Below are the replies to the reviewer's comments, and indications of additions, modifications or subtractions to the text under discussion. We report the reviewer's comments in italics, our responses in roman, and the text added to the manuscript in roman blue.

- *The discussion of instruments covers pages 5-8, and the analysis methods between 9 and 12, which I think is excessive, given that the written text is 20 pages in total. And there is virtually no discussion of the clouds sampled-the article is more of an instrument comparison study. There's a little discussion (a few sentences) just before the conclusions section but should have much more detail earlier in the manuscript. A discussion of earlier measurements in similar types of cloud conditions is warranted.*

In the revised manuscript we have tried to reduce the discussion of instruments and method of analysis. In fact, the manuscript was written with the intention of limiting ourselves to an instrumental comparison, and not to an in-depth discussion of the type of clouds encountered and the morphologies of their cloud particles. That can be found in other papers presented in the special issue *StratoClim stratospheric and upper tropospheric processes for better climate predictions* (ACP/AMT inter-journal) (Krämer et al. 2020, Lamraoui, F. et al, 2022; Khaykin et al., 2022). References to previous measures were inserted in the text (see lines 57-63) and in particular two articles (Baumgardner et al., 2017; Kramer et al, 2020) are an in-depth review of previous measurements in cirrus.  We agree on the need for a description of the type of clouds observed, for this reason we have inserted in the revision of the manuscript, following line 76, the following text:

Most of the measurements during StratoClim were performed at temperatures ≲ 205 K, corresponding to potential temperatures ≳ 355 K and altitudes ≳ 14 km, i.e., in the TTL. Krämer et al. (2020) reports a description of the clouds observed during the campaign. The first part of the campaign period suffered from very rare cloud passes at elevated altitudes, with comparatively low cloud particle number concentrations (below 1 cm$^{-3}$). In fact, during this period, the vast majority of clouds were encountered during ascent from or during approach to the Kathmandu airport. On 29 and 31 July, and on 02 August,

most of the clouds were come across at pressure level of ~400 hPa (and higher) during ascent and descent, with cloud particle concentrations ranging from 100 to 1000 cm$^{-3}$. The second campaign period (flights on 04, 06, 08, and 10 August 2017) provided extended fields of cirrus clouds of convective origin, with elevated particle densities and broad size distributions covering almost the entire detection size range of the different particle probes. The cloud particle measurements, mostly carried out over the southern slopes of the Himalayas captured high ice water content up to 2400 ppmv and ice particle aggregates exceeding 700 µm in size. The observed ice particles were mainly of liquid origin, with only a small amount formed in situ. ERA5 reanalysis corroborates the presence of high IWC detrained from deep-convective. A microphysical modeling study by Lamraoui et al. (2023) focuses on the flight of the 10 August, but its results can probably also be applied to the other cases of convectively generated cirrus measured during the second part of the campaign. The study predicts ice habits and reproduces the observed IWC, ice number concentration, and bimodal ice particle size distribution. The lower range of particle sizes is mostly represented by planar and columnar habits, while the upper range is dominated by aggregates with sizes between 600 and 800 µm. The study suggests that most of measured ice particles are of liquid origin with only a small amount formed in situ. These latter are associated with low values of IWC and number concentration, which makes them less influential in regulating the IWC which is, on the contrary, substantially influenced by planar ice particles of liquid origin. The difference in ice number concentration across habits can be up to 4 orders of magnitude, with aggregates occurring in much smaller numbers.

- *2. There are numerous acronyms used throughout the article. A table of the names associated with acronyms is warranted.*

A table of acronyms has been added to the revised manuscript.

- *3. Another point: Given that your PSD instruments only measured up to 960 microns yet you are sampling convective outflow cirrus, is this a problem? Do you have any particle that are in the largest size bin?*

There are indeed occasions when even the largest dimensional bins reveal the presence of particles. However, these occurrences do not seem to significantly influence the calculation of the backscatter ratio, as shown in Figure 16 which illustrates the buildup of the particle backscattering coefficient with respect to the particle radius, for all the particle size distribution under study. From this figure it can be seen that is mainly the dimensional range between 100 and 300 µm in radius that influences the final value of the backscattering. A similar consideration can also apply to the Surface Area Density of the particulate matter. We also performed similar calculations for the Volume Density, arriving at similar results. We can therefore state that the main contribution to the bulk microphysical parameters of the cloud comes from particles between 200 and 400 µm in radius. We recall here that the estimate of the IWC presented in our work mainly derives from hygrometric measurements, and only on two occasions was it estimated from the size distribution of the particulate matter (line 341). We also noted in the manuscript that an extensive assessment of the methodology to retrieve IWC from the two hygrometers and from the PSDs, as well as a comparison between the two approaches, has been carried out in Afchine et al. (2018) who demonstrated the equivalence of the two methods.

Furthermore, from PSD presented in the literature (e.g. Tian et al, 2010; Lawson et al., 2019) we expect a reduction of 2-4 orders of magnitude in the concentration of particles larger than 500 µm in radius, compared to those in the range 100-300 mm in radius.

Finally, A PIP also was concurrently operated with the other cloud probes. The data demonstrate that the number densities for particles larger than 960 µm were often four orders of magnitude or

more below the concentrations reported by other instruments, with only very few occasions where this ratio was reduced to three orders of magnitude. We are therefore confident of the relative non-influence of the particulate with linear dimensions greater than one μm, in determining the value of the backscattering. An extremely conservative estimate of the contribution to the backscattering coming from the particulate with a diameter greater than 960 μm does not exceed a few percentage units in rare worst cases. So, in order to keep the computations manageable and the paper compact, we stayed away from introducing the PIP instrument together with the corresponding measurements.

- *4. Lines 398-400. It's a very smart idea to compare backscattersonde to the lidar data*

We thank the reviewer for this kind remark.

- *5. Figure 4. Can you change the units to something that might be more meteorologically oriented, for example m-1? This is very much the case for Figure 7-11, and ones that follow.*

There is no general consensus on how to represent the measurement units of the backscattering coefficient, which in lidar practice is indifferently indicated in $m^{-1}$ $sr^{-1}$, $km^{-1}sr^{-1}$ and also, albeit less frequently, in $Mm^{-1}$ $sr^{-1}$. We will comply with the reviewer's suggestion by indicating it in $m^{-1}$ $sr^{-1}$ here and elsewhere.

- *6. Section 4.3, Figures 7-1q. I feel strongly that the units for backscattering coefficient in the figures km-1 sr-1 should be something that modelers, etc could use. These should be put in standard cloud physics units. Also, IWC should be in g/m3. This would facilitate comparison with other studies (for example, IWC in Figure 11 to Thornberry Thornberry, T. D., A. W. Rollins, M. A. Avery, S. Woods, R. P. Lawson,T. V. Bui, and R.-S. Gao (2017), Ice water content-extinction relationships and effective diameter for TTL cirrus derived from in situ measurements during ATTREX 2014, J. Geophys. Res. Atmos., 122, 4494–4507, doi:10.1002/.*

We now comply with the reviewer's suggestion by indicating in all our figures the backscattering coefficient in $m^{-1}$ $sr^{-1}$, and IWC in $g$ $m^{-3}$. We thank the reviewer for pointing out Thornberry et al., 2017. This has led us to a revision of the way we present the comparison of IWC determinations with respect to the extinction. Figure A9 has been suppressed while a new comparison figure has been inserted in the text, with several IWC-σ regressions, illustrated in a new table.

[Figure]

Figure 18: Scatterplot of measured IWC vs estimated extinction s=30b. The solid lines represent regressions from i. the present work, black; ii. Heymsfield et al. (2005), purple; iii. Avery et al., (2012), brown; Heymsfield et al. (2014), (a) yellow; (b) green; Thornberry et al. (2017), blue. Experimental points are color-coded in temperature of the observation.

| IWC[g m⁻³] - σ[m⁻¹] parametrization | | |
|---|---|---|
| Reference | Functional form | T range |
| Heymsfield et al. 2005 | IWC=119*$\sigma^{1.22}$ | 198-263 K |
| Avery et al., 2012 | IWC=238*$\sigma^{1.22}$ | |
| Heymsfield et al. 2014 (a) | IWC=a*$\sigma^b$
a=0.00532*( T[°C]-183)$^{2.55}$
b=1.31*exp(0.0047*(T-273)) | 188-270 K |
| Heymsfield et al. 2014 (b) | IWC=$\sigma$(0.91/3)*91744*exp(0.177*(T-273))
IWC=$\sigma$(0.91/3)*83.3*exp(0.0184*(T-273)) | 202-217 K
188-202K |
| Thornberry et al., 2017 | IWC=$\sigma$*(0.92/3)*(40+0.53*(T-192))
IWC=$\sigma$*(0.92/3)*(12+28*exp(0.65*(T-192))) | 192-207 K
185-192 K |
| Present work | IWC=1552*$\sigma^{1.39}$ | |

Table 2: IWC-σ Parameterizations (adapted from Thornburry et al., 2017)

To update the text and present the figure, lines 571-584 have been deleted and the following text has been inserted:

Several studies have provided an estimate of the dependence of the IWC on lidar extinction (Heymsfield et al.; 2005, Avery et al., 2012; Heymsfield et al, 2014; Thornberry et al., 2017). They are based on in situ measurements of IWC and PSD, the latter used to provide an estimate of the lidar extinction from optical modeling of the cloud particles.

These IWC-σ relationships could be compared with our IWC estimates based on β, if a suitable extinction-to-backscatter ratio (a.k.a. Lidar Ratio) LR is chosen. Unfortunately, LR can vary from 10 to 40 sr in tropical cirrus clouds (Chen et al., 2002) thus making the comparison somewhat arbitrary. Using a LR = 30 sr as a

most probable value (Balmes etal., 2019) and posing σ = LR*β we can correlate our IWC measurements with σ. Figure 16 is therefore the analogue of figure 11, where this time IWC is reparametrized as a function of σ.

The same figure shows the analytical relationship obtained in this work (solid black line), with those present in the literature, shown in table 2. Although all parameterizations capture the IWC-σ trend, and align with each other in the lower range of data variability, the result of our study is in agreement only with Avery et al. (2012) while it diverges from the other parameterizations, more severely for those that depend on the temperature. This especially in the upper range of data variability. It should be noted that in this range, the data themselves have also a greater dispersion.

We want to underline the limits of this comparison: in the case of the present study, they are caused by having chosen, rather arbitrarily, the same LR value for all the clouds observed, and in the case of the other parametrizations they are caused by having used an indirect determination of the extinction, calculated from the PSDs. It would be very interesting to have simultaneous in situ observations of backscattering, extinction and IWC available in the future.

- *7. Lines 524-525. The greatest ambiguity in the results of the comparison is linked to the choice of particles' morphology. Perhaps you should temper this because the mass dimensional relationship and the masses of the small particles are not compared directly to FLASH.*

There is an ambiguity due to an imprecise formulation of our sentence, which in effect only comments on the part of the study involved in the comparison of the observed versus the calculated backscattering. We have rephrased the sentence (524-525) as follows:

… it is clear from our simulations that the greatest ambiguity in the results of the observed versus the calculated comparison is linked to…

- *8. The relative independence of $\beta$ from $R_{mean}$ and $R_{eff}$ confirms $N_{ice}$ as the main parameter governing the cirrus scattering properties at optical wavelengths. Does this result also fall out of the analytical relationships assuming gamma PSD and quasi-spherical ice particles?*

Gamma distributions can be used to represent the number distribution function n(r), often used in GCMs to for the size distribution for each class of hydro-meteors in the model (e.g., ice, snow, and rain) (Gettelman et al., 2010) and it is given by:

$$n(r) = ar^{\alpha}e^{-br}$$

This distribution has a mode radius of $r_m = \frac{\alpha}{b}$, while the particle concentration is expressed by $N_0 = ab^{-\alpha-1}\alpha!$ (Grainger, 2017).

In our study we demonstrated how most of the backscattering is built up in the particle range 100-300 μm (see fig. 16), where the size parameter $\frac{2\pi r}{\lambda}$ of the particle is much greater than 10 and this result suggests to place oneself in the geometrical optics approximation, setting the backscattering efficiency to be unitary, so as to place in the calculation of the backscattering coefficient:

$$\beta \cong \int_0^{\infty} \pi r^2 n(r)\, dr$$

(this is the integral form of our eq. (11)). Moreover, when α is an integer, the moments of the gamma distribution have a simple analytical expression:

$$m_i = ab^{-\alpha-1-i}\Gamma(\alpha + i + 1) = ab^{-\alpha-1-i}(\alpha + 1)!$$

so as to have:

$$\beta \cong \pi a b^{-\alpha-1} \alpha! \, b^{-2}(\alpha+1)(\alpha+2) = N_0 \pi b^{-2}(\alpha+1)(\alpha+2) \sim N_0 \left(\frac{\alpha}{b}\right)^2 = N_0 r_m^2$$

So $\beta$ turns out to be linear in $N_0$, but also approximately quadratic in $r_m$.

We arrive at a similar result using a lognormal instead of the gamma distribution for the PSD

$$n(r) = \frac{N_0}{\sqrt{2\pi}} \frac{1}{r \ln(s)} e^{\left[-\frac{(\ln(r) - \ln(r_m))^2}{2 ln^2(s)}\right]}$$

with identical approximations:

$$\beta \cong \pi N_0 r_m^2 e^{2 ln^2(s)} \sim N_0 r_m^2$$

(Grainger, 2017). Thus a dependence on the square of the modal radius – and hence of other similar parameters linked to it, as the mean or the effective radius - is indeed to be expected, as the physical intuition would also suggest. In our case, the variability of such radius is of the order of a factor 2 (refer to figure 8 and 9) while the variability of the particle concentration extends for five orders of magnitude, and for this reason it turns out to be the main factor driving the backscattering variability.

To clarify this point, we have deleted lines 443-445 and inserted the following text:

With simple analytical calculations on various types of functional forms for the PSD (gamma, lognormal, etc.), and in the spherical ice approximation, it is easy to demonstrate that a dependence on the square of the modal radius – and hence of other similar parameters linked to it, as the mean or the effective radius - as well as by the total number of particles, is indeed to be expected for b, i.e. $\beta \sim N_0 r_m^2$ , as the physical intuition would also suggest. In our case, such a dependency on $r_m$, which varies by a factor of 2, is masked by the much wider variability of $N_0$, which varies over five orders of magnitude.

***Minor Comments:***

*I have numerous minor comments that should be considered in the revision of the manuscript.*

*Line 3. in view to > with the goal of connecting*

Corrected in the new version of the manuscript.

*7. Hymalaian to Himalayan*

Corrected in the new version of the manuscript.

*14? What do you mean by "can be set"*

The computed backscattering coefficient can be brought into good agreement…

*26: Cirrus at higher altitudes. Regarding your statements about cirrus, it would be good to use the AMS Glossary of Meteorology definition.*

Lines 25-26 have been substituted with:

Cirrus are high clouds existing between -35° and -85°C, composed of ice crystals, of micron to millimeter size (Lynch et al. 2002), that are fairly widely dispersed, usually resulting in relative transparency and whiteness and often producing halo phenomena not observed with other cloud forms (AMS, 2023).

*85. properties. This is particularly…*

Corrected in the new version of the manuscript.

*98. Why not use the Self-Similar Rayleigh-Gans Approximation (Hogan and Westbrook, 2014), or DDSCAT?*

As far as we know, the Self-Similar Rayleigh-Gans Approximation (SSRGA) is mostly used to compute the scattering from aggregated ice particles and snowflakes in the microwave and millimeter parts of the spectrum. For such kind of particle shapes the soft sphere/spheroid approximation tends to significantly underestimate scattering. In our case, given the supposed prevailing morphology of the diffusers and the wavelength used, we preferred a different approach for the scattering.

Discrete Dipole Approximation (DDA) could have been an adequate choice, and DDSCAT a good implementation of it. Indeed, the program supports calculations for a variety of target geometries (e.g., ellipsoids, regular tetrahedra, rectangular solids, finite cylinders, hexagonal prisms, etc.). However, is very computationally costly, and it is reported a lack of convergence for size parameters > 25 (Draine and Flatau, 2013)., much smaller than those encountered in our study.

*118. developed by*

Corrected in the new version of the manuscript.

*265. This line should be part of paragraph on line 264.*

Corrected in the new version of the manuscript.

*285-330 Very nice, comprehensive calculations of how the aspect ratio affects the*

*backscattering efficiencies.*

We thank the reviewer for this kind remark.

*339. Could you include the m-D relation you use in the text?*

Added in the new version of the manuscript (line 339):

… we have used the m–D relation described in Krämer et al. (2016): $m=a*D^b$ where a = 0.001902, b = 1.802 for D > 240 µm; a = 0.058000, b = 2.700 for D = 10–240 µm; ice crystals are spheres for D < 10 µm. The validity of the m-D relation is verified by Afchine et al. (2018) by comparison to thirteen others.

*368-370 Very good determination of why the NIXE-CAPS data set was used.*

We thank the reviewer for this kind remark.

*Section 4.2. It's clever to use the backscatter model together with the measurements for find*

*the best AR.*

We thank the reviewer for this kind remark.

*385. Remove i*

Corrected in the new version of the manuscript.

*405. backscattering should not be capitalized*

Corrected in the new version of the manuscript.

*408. range is*

Corrected in the new version of the manuscript.

*440. as seems to do > as well as SAD and IWC*

Corrected in the new version of the manuscript.

*441. as hardly change. Please rephrase*

Corrected in the new version of the manuscript.

*475 and and*

Corrected in the new version of the manuscript.

*512. We remind. We note that...*

Corrected in the new version of the manuscript.

Bibliography (related to the present answer and not previously quoted in the manuscript, new addition in the manuscript are in blue):

American Meteorological Society, 2023: "cirrus", "cirriform". Glossary of Meteorology, http://glossary.ametsoc.org/wiki/cirrus , http://glossary.ametsoc.org/wiki/cirriform

Avery, M., D. Winker, A. Heymsfield, M. Vaughan, S. Young, Y. Hu, and C. Trepte (2012), Cloud ice water content retrieved from the CALIOP space-based lidar, Geophys. Res. Lett., 39, L05808, doi:10.1029/2011GL05045.

Draine B.T and Flatau P. J., User Guide for the Discrete Dipole Approximation Code DDSCAT 7.3, arXiv:1305.6497 [physics.comp-ph]

Gettelman, A., X. Liu, S. J. Ghan, H. Morrison, S. Park, A. J. Conley, S. A. Klein, J. Boyle, D. L. Mitchell, and J.-L. F. Li (2010), Global simulations of ice nucleation and ice supersaturation with an improved cloud scheme in the Community Atmosphere Model, J. Geophys. Res.,115, D18216, doi:10.1029/2009JD013797.

Grainger, R. G.: Some Useful Formulae for Aerosol Size Distributions and Optical Properties, 2022, accessed May 15, 2023 at: http://eodg.atm.ox.ac.uk/user/grainger/research/aerosols.pdf

Lamraoui, F., Krämer, M., Afchine, A., Sokol, A. B., Khaykin, S., Pandey, A., and Kuang, Z.: Sensitivity of convectively driven tropical tropopause cirrus properties to ice habits in high-resolution simulations, Atmos. Chem. Phys., 23, 2393–2419, https://doi.org/10.5194/acp-23-2393-2023, 2023.

Thornberry, T. D., A. W. Rollins, M. A. Avery, S. Woods, R. P. Lawson,T. V. Bui, and R.-S. Gao (2017), Ice water content-extinction relationships and effective diameter for TTL cirrus derived from in situ measurements during ATTREX 2014, J. Geophys. Res. Atmos., 122, 4494–4507, doi:10.1002/.

Tian, L., G. M. Heymsfield, L. Li, A. J. Heymsfield, A. Bansemer, C. H. Twohy, and R. C. Srivastava, 2010: A Study of Cirrus Ice Particle Size Distribution Using TC4 Observations. J. Atmos. Sci., 67, 195–216, https://doi.org/10.1175/2009JAS3114.1.

Lawson, R. P., Woods, S., Jensen, E., Erfani, E., Gurganus, C., Gallagher, M., et al. (2019). A review of ice particle shapes in cirrus formed in situ and in anvils. Journal of Geophysical Research: Atmospheres, 124, 10049– 10090. https://doi.org/10.1029/2018JD030122

Draine, B. T., Flatau, P. J., Discrete-dipole approximation for periodic targets: theory and tests,

Gettelman, A., X. Liu, S. J. Ghan, H. Morrison, S. Park, A. J. Conley, S. A. Klein, J. Boyle, D. L. Mitchell, and J.-L. F. Li (2010), Global simulations of ice nucleation and ice supersaturation with an improved cloud scheme in the Community Atmosphere Model, J. Geophys. Res.,115, D18216, doi:10.1029/2009JD013797.

**Answers to the Second Reviewer's Review of "A Comparative Analysis of In-Situ Measurements of High Altitude Cirrus in the Tropics", by Cairo et al., MS No.: egusphere-2023-112.**

The Authors thank the Reviewer for the kind benevolence towards the manuscript and for its careful analysis which improved the manuscript and has also stimulated many new ideas for future work. Below are the replies to the reviewer's comments, and indications of additions, modifications or subtractions to the text under discussion. We report the reviewer's comments in italics, our responses in roman, and the text added to the manuscript in roman blue.

**Comments**

- *l. 90 - "modelling of the optical properties of cirrus clouds is a formidable problem..." I share the authors' pain. The rest of the paragraph provides a nice review of previous efforts tackling this problem, but I'm not sure it brings a lot to the current discussion.*

We would like to maintain the review to document the possible different approaches to modeling aspheric particulate scattering.

- *section 2 -- the number of instruments is a bit large, and a table or a figure summarizing them would be nice.*

The revised manuscript will have the following table:

| Instrument | Parameter measured | Technique | Range | Sensitivity | Resolution |
|---|---|---|---|---|---|
| CCP | Cloud PSD | Laser-optical particle counter | Particle diameter: 3-47 um concentration<2000 cm-3 | single particle detection | 1 s |
| CIP | Cloud PSD | Laser-optical Cloud particle Imaging | Particle diameter: 25-1550 um, concentration < 500 cm-3 | single particle detection | 1 s |
| NIXE-CAPS | Cloud PSD and shape | Laser-Particle spectrometer | Particle diameter 0.61 - 937 µm | Single Cloud Particle detection | 1 s |
| MAS | particle backscattering and depolarization | Laser elastic scattering | backscatter coefficient: 5x10−7 - 10-1 km−1 sr−1 volume depolarization 0-100 (%) | Backscatter coefficient 5x10-7 km−1 sr−1 volume depolarization 1 (%) | 10 s |
| FISH | H2O (total) | Lyman-a | 1-1000 ppmv | 0.1ppmv | 1 s |
| FLASH | H2O (gas phase) | Lyman-a | 1 – 1000 ppmv | 0.1 ppmv | 1 s |

- *l. 132 - "the resolution is 10 s" -- I was surprised by this (relatively) very long sampling interval. It is worse than the 5 s mentioned in Cairo and al 2011. This means the MAS samples one point every 1 or 2km, which is quite a large distance. Are there ways to improve this?*

Unfortunately, not. The wear of the laser, no longer in production, has led to a reduction in the power emitted during the course of the lifetime. If we will have the opportunity to rebuilt the instrument, we will improve the temporal resolution.

- *Figure 1 (1) : it is unclear to me why in Figure 1 the highest concentrations of points are not located at BR=1 ? Since clouds show up on 7 hours from the entire 35 hours of measurement, I would expect most of the points sampled by the MAS to be cloud-free, and thus to produce a BR=1 measurement. Could you please clarify my misunderstanding?*

This observation is correct. In fact, we did not specify in the Figure 1 caption that only the points were represented for which it was possible to calculate the Total Particle Depolarization with satisfactory S/N, in order to maintain a consistency of the dataset between figures 1 and 2. For BR close to 1 the aerosol depolarization has a too large uncertainty, or lacks sense as no particles are present. This excluded many points close to BR=1. But in fact this is not strictly necessary, therefore we will replace Figure 1 with the one below, which includes all the observations, and update the captions to Figure 1 and 2, and the text accordingly.

[Figure]

- *Figure 1 (2) : could you specify in the legend the total number of data points?*

Figure 1. 2D histogram of Backscatter Ratio vs altitude. Data were acquired throughout the campaign by the MAS backscattersonde. The color codes the number of observations in the 2D bin, representing 8477 data points.

Figure 2. 2D histogram of Total Particle Depolarization data vs temperature for altitudes above 11 km. Data were acquired throughout the campaign by the MAS backscattersonde. The color codes the number of observations in the 2D bin, representing 2308 data point.

After line 160: To avoid contamination by aerosols or an excessive uncertainty on the depolarization for very low BR, we will restrict our analysis to observations with BR>1.2, for a total of 2132 data points.

- *l. 158: "a negative trend with respect to temperature can be discerned" this is also reportedly the case in CALIPSO data, Sassen et al 2009 -- but I agree this trend is frustratingly hard to document quantitatively.*

We thank the reviewer for pointing out the Sassen et al. (2009) article. A reference to it has been added in the text (line 159)

- *In Methods : couldn't the CCP probes provide info about the particle aspect ratio? Such info could then be compared to the AR retrieved from the beta fit. Or do you think the aspect ratios from probes are too uncertain?*

The aspect ratio is not a standard output of the CCPs and their data have not been processed at this level of detail, so in our work we restricted ourselves to a comparison between the T-Matrix derived Aspect Ratio and the Aspherical Fraction of NIXE-CAPS. Even if we could get aspect ratio out of the CIP data – they could be biased (see what is written in line 364 and following): "This resulted in a slightly lower instrument particle detection sensitivity [Ed., for the CCP] which could be only identified by a comparison of particle habits like size and number concentrations in the range of both CIPgs instruments measured at significantly elevated altitudes (Port, 2021).". Moreover, in Weigel et al. (2016), something similar was performed for spherical drops under known conditions to determine poor timing of CCP sampling frequency. It became clear that AR in CCP images is easily variable due to airspeed errors. For ice particles (3D objects projected onto a 2D plane) it could be very easy to obtain values from the images, but whether they really represent AR is a matter of believing or not. So due to the technical limits of the CCP we agreed that the uncertainty would be too high to trust kind of AR values that were extracted in this way from the CCP data.

- *l. 250: there is something wrong with the URL, I'm guessing the underscores were swallowed by the formatting system.*

The correct URL is https://oceancolor.gsfc.nasa.gov/docs/ocssw/bhmie_8py_source.html

We have corrected that in the manuscript.

- *l. 255: clicking the URL here brought me to the American Express website.*

That is bizarre. The URL is in fact https://www.grasp-open.com/

- *l. 263-266: there is a problem with formatting here*

Corrected in the new version of the manuscript.

- *l. 272: missing period*

Corrected in the new version of the manuscript.

- *l. 371: this paragraph suddenly mentions beta NC. I think this is the first time in the paper that beta NC is mentioned, and I don't think it is defined. Is beta NC the best match for the measured beta among the simulated beta AR NC ?*

Correct. In the new manuscript, at line 371: we select point-by-point the $\beta^{AR}_{NC}$ that provides the best match with the $\beta$ measured by the backscattersonde, and let $\beta_{NC}$ be this best match.

- *Figure 3 (1): have you tried to create a scatterplot of depolarization vs AF? vs AR? of AF vs AR? Could you perhaps write the various labels of the vertical axis in the same color as their associated plot? It would make it easier to parse the figure.*

Depolarization vs AF is reported in the supplementary material as "Figure A8. 2D histogram of Particle Aspherical Fraction (y-axis) vs Particle depolarization (x-axis)". We are afraid it does not tell much.

The plot of AF vs AR does not give interpretable results.

Perhaps more of interest to look at depolarization vs AR, hereby presented, color coded in terms of the backscattering coefficient.

[Figure]

Here we seem to discern a tendency to have AR not far from 1 associated with small $\beta$ and low depolarization, while AR >1.5 and AR<0.5 tend to be more present at depolarization >20%, associated with medium-high $\beta$.

This can be quoted in the manuscript, and what shown above added as a new Figure A6 in the supplementary material. We propose to add at line 393: Looking at the entire dataset, we note a tendency to have ARs close to 1 associated with small β values and depolarization of less than 20%, while ARs greater than 1.5 and less than 0.5 favors medium-high β values and depolarization greater than 20% (see Figure A6 in the supplementary material)

- *Figure 3 (2): is the AR retrieval at each time stamp completely independent from the previous retrievals? In some parts of the plot it looks like the retrievals are "stuck" at AR=3.*

Yes, each time stamp is independent from the past. Our interpretation of this is that the best AR selection "sticks" to AR=3 in regions of the clouds when the calculated backscattering is at its lowest value, but still too high to match the observations. In fact, from figure A1 it can be seen that AR=3 is the choice that gives the smallest backscattering, among the possible choices of AR.

- *Figure 3 (3): Figure 3, like most of the figures, is a bit on the small / crude side. This makes reading the figure harder than it should. Could you please produce higher-res versions of figures next time?*

Corrected in the new version of the manuscript.

- *Figure 3: I understand this is a bit outside the scope of the paper, but it would be very interesting to document the spatial scales over which depolarization is homogeneous.*

This is an interesting point, which would be worth exploring with the entire MAS dataset from all the campaigns in which it participated. The work plan could be to select continuous time series - possibly on airplane trajectories with constant height and heading - in which clouds have been observed, and to find a metric of homogeneity, for example the variance of the residuals with respect to a running average. We intend to carry out this study in detail, and this is also planned for a future work focusing on the 10 August 2017 flight. We are afraid that performing this analysis in this present context would expand our manuscript too much. However, for the sake of completeness, we are presenting here an attempt in such direction. In the following figure, a portion of the 10 August 2017 flight, the particle depolarization is shown (red dots) together with a running standard deviation (STD) of its residuals (blue dots). Residuals have been computed by subtracting to the depolarization its running average computed over 50s. Running STD has been computed as a moving STD over a 50s window. In a future work we will try to fix the many loose ends of this analysis (dependence on the running average window length, on the aircraft trajectory and so on…) and to connect the values of the SDT to the dynamical state of the atmosphere, by inspecting its relationship with the variability of winds and temperatures and, perhaps, supersaturation.

[Figure]

Figure caption: Red dots, particle depolarization measured by MAS in a portion of the 10 August 2017 flight; Blue dots, running Standard Deviation of the residuals of the particle depolarization.

- *l. 400: "(19000s and 21000s)" what are those numbers? Can they be found somewhere on a figure? On Figure 3 times only go from 33000 to 38000.*

There, we are mentioning which part of the dataset was used for the MAS-MAL comparison. The data for 10 August are shown in Figure 3, those for 8 August should have been shown in figure A6 in Supplementary Material, which however is wrong in the manuscript, being at present Figure A6 a replica of Figure 3. We apologize for this. The corrected Figure A6, shown below, will be inserted correctly in the revised manuscript.

[Figure]

Figure A6. Red line, time series of particle backscattering coefficient $\beta$ measured by MAS on 8 August 2017; Black solid line, $\beta_{NC}$ corresponding to the best matching between measured $\beta$ and computed $\beta^{AR}_{NC}$ (not displayed), error bars are reported in the first part of the curve; black dashed lines, maximum and minimum values of the optical modeling $\beta^{AR}_{NC}$ values. Lower panel: Red line , particle depolarization measured by MAS; Blue line Particle aspherical fraction measured by NIXE-CAPS; black line, AF values of the best matching between measured $\beta$ and computed $\beta^{AR}_{NC}$.

- *l. 400-405: Are the MAS/MAL depolarization ratios similarly correlated? Does the results from figure 4 suggest it is possible to use MAL measurements instead of MAS when more practical?*

In fact, there is a similar agreement, albeit more scattered, between depolarization from MAL up/down and MAS. Thus it is probable that the agreement is conserved in all regions of the clouds not close to their edges where strong gradients should be expected, and therefore it would be possible in general to use and process the lidars MAL up/down, in the partial overlap region very close to the plane, as backscattersondes. Then we would have, at least approximately, the in-situ optical parameters of the cloud. However, it should be remembered that the time resolution of the MAL lidars is 1 minute.

- *Figure 4: Using a square aspect ratio in Figure 4 would make sense.*

Corrected in the new version of the manuscript.

- *Figure 5 and 6: Having both as subplots of the same figure would help the reader. In Figure 5 I find it confusing that the y-axis says "Backscatter coefficient NIXE-CAPS" -- the NIXE-CAPS does not measure the backscatter coefficient. Same for figure 6.*

In the revision of the manuscript we will combine the two figures. Here and where necessary we will change "Backscatter coefficient NIXE-CAPS" to "Backscatter coefficient (computed from NIXE-CAPS)"

- *l. 419: "From the inspection of Figure 6 we can see an attempt to achieve a 1-1 correlation between the two datasets". I don't understand this sentence.*

Rephrased as (419): From the inspection of Figure 6 we see that there are few cases when the two depolarization line up to a 1-1 correlation. However, for a considerable number of observations, the calculated depolarization remains around 10% while the measured one varies throughout its range of variability.

- *l. 421: "the optical modelling completely fails" -- completely is perhaps a strong word, sometimes the retrieval is correct. But I agree it's perhaps worse than chance, the performance is very poor.*

We have rephrased it as (421): the optical modelling performance is very poor in reproducing…

- *l. 444 : "most of them are so called irregulars" how do you know this? Is this based on CCP imagery?*

No, it is rather a conjecture to suggest that in cirrus clouds formed in-situ, "irregular" particles are less frequent (Lawson et al., 2019) However, in this second reading, we would prefer to delete the sentence, also in view of the fact that, following suggestions by the first reviewer, we replaced lines 443-445 with: With simple analytical calculations on various types of functional forms for the PSD (gamma, lognormal, etc.), and in the spherical ice approximation, it is easy to demonstrate that a dependence on the square of the modal radius – and hence of other similar parameters linked to it, as

the mean or the effective radius - as well as by the total number of particles, is indeed to be expected for $\beta$, i.e. $\beta \sim N_0 r_m^2$ , as the physical intuition would also suggest. In our case, such a dependency on $r_m$, which varies by a factor of 2, is masked by the much wider variability of $N_0$, which varies over five orders of magnitude.

- *l. 476: "We remind again that" perhaps with some editing it would be possible to avoid two reminders*

Corrected in the new version of the manuscript.

- *l. 477-485: This discussion is great. Based on Figure 12, wouldn't it be possible to combine the measured volume depolarization ratio and BR to retrieve the particulate/aerosol depolarization ratio (by extending the "line" these two measurements fall on all the way to the right), and use it instead of the volume depolarization ratio throughout your study? You might find better correlation with the AR time series.*

Thank you for such remark. In fact, in our study we have always used the particulate/aerosol depolarization ratios $\delta_A$ and $\delta_{TA}$, as defined in formulae (3) and (4) and computed from the volume depolarization ratio and the backscatter ratio as in Cairo et al, (1999). Only in figure 13-15 we have reported $\delta_T$ (and intuitively, on extrapolation, $\delta_{TA}$) as we think the Adachi et al. (2001) way of displaying depolarization is a nice way of displaying at a glance the whole dataset, and investigate its dependence on cloud parameters and on temperature.

- *l. 490-493: It is not clear to me what is learned from Figure 13. The strong correlation between BR and Nice has already been shown before.*

There is a bit of information that may be added to that. We have tried to make that clearer with this revised text. From line 491 on: Figure 13 clearly shows again the positive correlation between BR and Nice. This relationship is independent from the polarization $\delta_{TA}$ throughout the BR range, except at the highest BR values. There, we note how for the same extreme values of BR, low $\delta_{TA}$ are associated with low $N_{ice}$, and high $\delta_{TA}$ with high $N_{ice}$.

- *Figure 15 is fascinating. Could you comment on whether particle orientation could have an effect on the volume depolarization ratio measured by the MAS, considering the specific MAS setup? Electric charging linked to lightning activity could lead to vertically-oriented particles, that would show planar faces to the MAS. Do you have any idea if the variable dynamic conditions (lines 500-504) could be disambiguated somehow by observation?*

This is a very interesting point to explore. From the comparison of the depolarization data from MAS and MAL on the time periods performed for producing figure 4, which reports depolarization values of similar values, although scattered, it does not seem that particle orientation had played a role. This is most likely because of the dynamical situation of the atmosphere – we were flying in the outflow of deep cumulus clouds - which prevented orientation to settle in. However, we have not performed an extensive comparison of the depolarization from the two instruments throughout the dataset and this is certainly a topic to look into for future work, also in view on a paper specifically addressing the 10 August 2017 flight, which is in preparation.

- *l. 508: "a measured aspect ratio would be an obvious candidate" -- I was under the impression imagery from the CCP could provide this kind of retrieval.*

Please refer to what has been reported in the previous answer to: *In Methods : couldn't the CCP probes…*

- *l. 579: From the start, it is not clear to me what is to be gained from following this path. Could you expand a bit on what you hoped to achieve with this?*

At the time of writing, no other direct comparisons between β and IWC were available in the literature, so we tried to compare the result we presented with analogous comparisons between extinction σ and IWC. For this purpose, we transformed our β into σ, assuming the most probable extinction-to-backscatter ratio LR=30 sr$^{-1}$ (Balmes et al., 2019), and tried to compare our IWC estimate from β, with those from Heymsfield (2005, 2014) using in them a σ derived from our β. This is the comparison presented in Figure A9. However, following the first reviewer's review, we propose to expand this comparison, eliminating figure A9, and introducing directly in the body of the manuscript a new figure in place, a new table and a new text.

[Figure]

Figure 18: Scatterplot of measured IWC vs estimated extinction s=30b. The solid lines represent regressions from i. the present work, black; ii. Heymsfield et al. (2005), purple; iii. Avery et al., (2012), brown; Heymsfield et al. (2014),  (a) yellow; (b) green; Thornberry et al. (2017), blue. Experimental points are color-coded in temperature of the observation.

| IWC[g m$^{-3}$] - σ[m$^{-1}$] parametrization | | |
|---|---|---|
| Reference | Functional form | T range |
| Heymsfield et al. 2005 | IWC=119*σ$^{1.22}$ | 198-263 K |
| Avery et al., 2012 | IWC=238*σ$^{1.22}$ | |
| Heymsfield et al. 2014 (a) | IWC=a*σ$^{b}$
 a=0.00532*( T[°C]-183)$^{2.55}$ | 188-270 K |

| | | |
|---|---|---|
| | b=1.31*exp(0.0047*(T-273)) | |
| Heymsfield et al. 2014 (b) | IWC=σ(0.91/3)*91744*exp(0.177*(T-273)) | 202-217 K |
| | IWC=σ(0.91/3)*83.3*exp(0.0184*(T-273)) | 188-202K |
| Thornberry et al., 2017 | IWC=σ*(0.92/3)*(40+0.53*(T-192)) | 192-207 K |
| | IWC=σ*(0.92/3)*(12+28*exp(0.65*(T-192))) | 185-192 K |
| Present work | IWC=1552*σ$^{1.39}$ | |

Table 2: IWC-σ Parameterizations (adapted from Thornburry et al., 2017)

To update the text and present the figure, lines 571-584 have been deleted and the following text has been inserted:

Several studies have provided an estimate of the dependence of the IWC on lidar extinction (Heymsfield et al.; 2005, Avery et al., 2012; Heymsfield et al, 2014; Thornberry et al., 2017). They are based on in situ measurements of IWC and PSD, the latter used to provide an estimate of the lidar extinction from optical modeling of the cloud particles.

These IWC-σ relationships could be compared with our IWC estimates based on β, if a suitable extinction-to-backscatter ratio (a.k.a. Lidar Ratio) LR is chosen. Unfortunately, LR can vary from 10 to 40 sr in tropical cirrus clouds (Chen et al., 2002) thus making the comparison somewhat arbitrary. Using a LR = 30 sr a most probable value (Balmes etal., 2019) and posing σ = LR*β we can correlate our IWC measurements with σ. Figure 16 is therefore the analogue of figure 11, where this time IWC is reparametrized as a function of σ. The same figure shows the analytical relationship obtained in this work (solid black line), with those present in the literature, shown in table 2. Although all parameterizations capture the IWC-σ trend, and align with each other in the lower range of data variability, the result of our study is in agreement only with Avery et al. (2012), while it diverges from the other parameterizations, more severely for those that depend on the temperature. This especially in the upper range of data variability. It should be noted that in this range, the data themselves have also a greater dispersion.
We want to underline the limits of this comparison: in the case of the present study, they are caused by having chosen, arbitrarily, the same LR value for all the clouds observed, and in the case of the other parametrizations they are caused by having used an indirect determination of the extinction, calculated from the PSDs. It would be very interesting to have simultaneous in situ observations of backscattering, extinction and IWC available in the future.

For completeness, we are now aware of a recent paper which also reports a direct comparison between β and IWC (Wagner et al., 2022), but the β variability there do not overlap with ours, so we preferred not to compare our result with that new work in the present revision of the manuscript.

- *l. 590: "this may induce an unquantified bias in our presented statistics" -- I'm not sure I understand the point this paragraph is trying to make. Do you wish to state that the results you present here are not necessarily representative of all cirrus that can be observed from space? This could be said in a more straightforward way. Also, this is not a knock against your study -- 7 hours of in-situ sampling cannot be expected to cover the entire range of cloud variabilities.*

We have rephrased that as: As there might be differences in the microphysical properties of the cirrus depending on the formation process (Lawson et al., 2019; Krämer et al., 2020) especially in the initial stage of their life cycle, the results presented here are not necessarily representative of all cirrus that can be observed from space.
In addition, the hours of in-situ sampling we have used in our analysis are closer to 6 than to 7. We have corrected that in the text.

**Technical comments**

*l. 408: "rangeis"*

Corrected in the new version of the manuscript.

*l .441: "are hardly change"*

Corrected in the new version of the manuscript.

*l. 461: "as instance" -- for instance?*

Corrected in the new version of the manuscript.

*l. 475: "and and"*

Corrected in the new version of the manuscript.

*l. 537: "MAS,."*

Corrected in the new version of the manuscript.

*many references contain two URLs, one for the DOI (see next comment) and one that leads directly at the publisher's website. The DOI link should (in theory) always resolve to the publisher's website, so both are duplicates, and one could be omitted without loss.*

Corrected in the new version of the manuscript.

*many DOI links appear as https://doi.org//https://etc (i.e. https:// appears twice). Clicking them goes to the right place, so I'm guessing this is a display issue.*

Corrected in the new version of the manuscript.

Bibliography (related to the present answer and not previously quoted in the manuscript, new addition in the manuscript are in blue):

Lawson, R. P., Woods, S., Jensen, E.,Erfani, E., Gurganus, C., Gallagher, M.,et al. (2019). A review of ice particleshapes in cirrus formed in situ and inanvils.Journal of Geophysical Research:Atmospheres,124, 10,049–10,090.https://doi.org/10.1029/2018JD030122

Wagner, S. W. and Delene, D. J.: Technique for comparison of backscatter coefficients derived from in situ cloud probe measurements with concurrent airborne lidar, Atmos. Meas. Tech., 15, 6447–6466, https://doi.org/10.5194/amt-15-6447-2022, 2022.

Weigel, R., Spichtinger, P., Mahnke, C., Klingebiel, M., Afchine, A., Petzold, A., Krämer, M., Costa, A., Molleker, S., Reutter, P., Szakáll, M., Port, M., Grulich, L., Jurkat, T., Minikin, A., and Borrmann, S.:

Thermodynamic correction of particle concentrations measured by underwing probes on fast-flying aircraft, Atmos. Meas. Tech., 9, 5135–5162, https://doi.org/10.5194/amt-9-5135-2016, 2016.